# Tumor cell plasticity, heterogeneity, and resistance in crucial microenvironmental niches in glioma

Erik Jung [1,2], Matthias Osswald[1,2], Miriam Ratliff[2,3], Helin Dogan[4,5], Ruifan Xie[1,2], Sophie Weil[1,2], Dirk C. Hoffmann[1,2], Felix T. Kurz[6], Tobias Kessler [1,2], Sabine Heiland[6], Andreas von Deimling [4,5], Felix Sahm [4,5], Wolfgang Wick [1,2] & Frank Winkler [1,2 ✉]

Both the perivascular niche (PVN) and the integration into multicellular networks by tumor microtubes (TMs) have been associated with progression and resistance to therapies in glioblastoma, but their specific contribution remained unknown. By long-term tracking of tumor cell fate and dynamics in the live mouse brain, differential therapeutic responses in both niches are determined. Both the PVN, a preferential location of long-term quiescent glioma cells, and network integration facilitate resistance against cytotoxic effects of radiotherapy and chemotherapy—independently of each other, but with additive effects. Perivascular glioblastoma cells are particularly able to actively repair damage to tumor regions. Population of the PVN and resistance in it depend on proficient *NOTCH1* expression. In turn, *NOTCH1* downregulation induces resistant multicellular networks by TM extension. Our findings identify *NOTCH1* as a central switch between the PVN and network niche in glioma, and demonstrate robust cross-compensation when only one niche is targeted.

---

[1] Neurology Clinic and National Center for Tumor Diseases, University Hospital Heidelberg, Heidelberg, Germany. [2] Clinical Cooperation Unit Neurooncology, German Cancer Consortium (DKTK), German Cancer Research Center (DKFZ), Heidelberg, Germany. [3] Department of Neurosurgery, University Hospital Mannheim, University Heidelberg, Mannheim, Germany. [4] Department of Neuropathology, Institute of Pathology, Ruprecht-Karls University Heidelberg, Heidelberg, Germany. [5] Clinical Cooperation Unit Neuropathology, German Cancer Consortium (DKTK), German Cancer Research Center (DKFZ), Heidelberg, Germany. [6] Department of Neuroradiology, University Hospital Heidelberg, Heidelberg, Germany. ✉email: frank.winkler@med.uni-heidelberg.de

Malignant gliomas, including glioblastoma, are the most frequent and yet incurable primary brain tumors characterized by early infiltrative growth and high therapy resistance[1]. Apart from frank angiogenesis, the vasculature of the brain and of the tumor seem to play a special role in glioma biology: a subpopulation of stem-like cells was identified in brain tumors that is potentially responsible for their frequent treatment failure[2–5], and these stem-like cells can inhabit distinct perivascular niches (PVN)[6–10]. Importantly, those stem-like glioma cells share marker expression and molecular traits with neural stem cells (NSCs)[2,3], and NSCs also populate distinct neurogenic niches, which include PVN that provide a unique microenvironment for the preservation of the stem cell pool[11]. Targeting of the PVN thus emerged as a strategy to eradicate cancer stem-like cells in incurable gliomas, to improve their response to therapy and to prevent tumor recurrence[9,12], but at the same time raised the question of the collateral effects associated with it.

In line with the neurodevelopmental origins of glioma[13,14] and shared traits between neural progenitor cells and glioma cells, neurite-like cellular protrusions named tumor microtubes (TMs) have recently been characterized in gliomas[15,16]. TMs are leading structures during glioma cell invasion[15,17] and enable glioma cells to form functional, communicating multicellular networks that render network members resistant against cytotoxic therapies, likely due to improved cellular homeostasis[15,18].

All in all, while there is increasing evidence that interactions with pre-existing blood vessels in the brain are of importance for tumor progression and resistance in glioma, their exact cellular and molecular mechanisms and overall role in brain tumor biology are not sufficiently understood. By following patient-derived primary glioblastoma cells dynamically for extended periods of time with in vivo two photon microscopy, we here describe mechanisms of tumor progression that depend on interactions with brain microvessels and tumor cell networks, and provide the definitive confirmation that the NOTCH1-dependent PVN is a primary niche of resistance in brain tumors. We also demonstrate that NOTCH1 is an important modulator of TM-network formation, and interference with this pathway leads to a partial cross-compensation between these two prime niches of resistance in glioma.

## Results

**A perivascular niche for tumor cells in glioma.** First, we asked how tumor cells in patient samples from diffusely infiltrating gliomas are distributed in relation to blood vessels in the brain. Mutation-specific immunohistochemical stainings against IDH1-R132H were used, which allow specific detection of tumor cell structures vs. nonmalignant brain cells in IDH1 mutant human oligodendroglioma and astrocytoma samples[19,20]. As validated before[21], nestin staining allowed to detect tumor cells in IDH wild-type glioblastoma. Irrespective of the glioma subtype, only a minority of tumor cells closely associated with blood vessels in a distinct perivascular niche (Fig. 1a; arrowheads in Fig. 1b). The majority of tumor cells populated the parenchymal compartment distant from blood vessels (Fig. 1a, with representative examples given in Fig. 1b where parenchymal, non-perivascular positions are evident on the left side images, but also visible on the right side images). This phenotype and population of different compartments could also be recapitulated in a patient-derived xenograft model growing in the mouse brain (Fig. 1c shows a microregion with primarily parenchymal, Fig. 1d one with primarily perivascular positions). In both human (Fig. 1b, e) and mouse (Fig. 1c, d) tumor samples, many parenchymal but also some perivascular cells were clearly characterized by the

extension of TMs, which connected single glioma cells to a network of tumor cells, as described before[15,21,22]. This TM-connected tumor cell network becomes most evident in 3D reconstructions of thick sections from mouse (Fig. 1c) and man (Fig. 1e).

In principle, the study of individual glioblastoma cells in these two niches—their plasticity, heterogeneity, stroma interactions, contribution to tumor progression, and their response to therapies—would greatly benefit from the ability to dynamically long-term track individual tumor cells[23]. Thus, to investigate the relative impact and dynamic interplay of the PVN and TM network niches, we used the above mentioned mouse xenograft models that closely reflect the growth pattern of malignant gliomas in humans, combined with intravital two photon-microscopy that allows to follow tumor microregions over long periods (up to months) in the three-dimensional space of the brain in living mice[15,17,18,21]. With this methodology we first analyzed general characteristics of tumor cells in the different compartments before investigating their responses to therapies, and finally molecular drivers of niche formation.

**Long-term quiescence in the perivascular niche.** Long-term tracking of glioma cells in invasive tumor regions (Fig. 2a) allowed to identify a subset of tumor cells that did not change their position over days to weeks, although being located at the invasive front. A significant majority of those long-term quiescent cells were found in a strict PVN position (Fig. 2b), closely following the course of blood vessels by stretching out to an elongated shape. A fluorescent reporter for tumor cell nuclei allowed to follow mitotic events in those cells. Here, the adjacent daughter cells remained closely associated with the respective blood vessel after cell division (Fig. 2c). Likewise, in the tumor core which is characterized by a higher cellular density and dense TM-connected tumor cell networks (Fig. 2a), the majority of PVN cells remained resident over weeks, whereas many parenchymal cells were still migrating (Fig. 2d). Histological analyses revealed that perivascular cell clusters were located between astrocytic endfeet and endothelial cells of brain microvessels (Fig. 2e), thus integrating into structures of the blood-brain barrier. Although the tumor cell soma remained at the very same spot over weeks, these perivascular cells actively extended and retracted TMs, scanning the environment and transiently or permanently connecting with neighboring cells (Fig. 2f). Some resident cells were closely associated with remodeled vessels such as capillary loops or glomeruloid-like bodies, with some cells ultimately migrating away and others remaining resident (Fig. 2g). The majority colonized morphologically unaltered vessels though.

As stem-like, potentially slow-cycling tumor cells are believed to populate the PVN in glioma[9], we next wanted to elucidate if cells in the different compartments vary in their proliferative activity. Histological analyses revealed that the proliferation rate of perivascular cells was lower compared to parenchymal cells in our xenograft models (Fig. 3a, b). In vivo EdU incorporation confirmed that most tumor cells in S-phase are located in the parenchyma, whereas perivascular cells are slow-cycling (Fig. 3c, d). In addition, intravital microscopy was used to analyze cell and vessel associations in larger 3D volumes, and demonstrated that the normalized mitotic rate of perivascular cells was indeed lower compared to the parenchymal subpopulation (Fig. 3e). Likewise, in molecularly defined human glioma specimen, the majority of proliferating, ki67-positive cells were located in the parenchymal compartment (Fig. 3f, g). The comparison of the distribution of tumor cells in the different compartments (Fig. 1a) and the proliferating fraction (Fig. 3f) suggests that perivascular tumor cells less actively divide in IDH1 mutant astrocytoma ($p = 0.0002$),

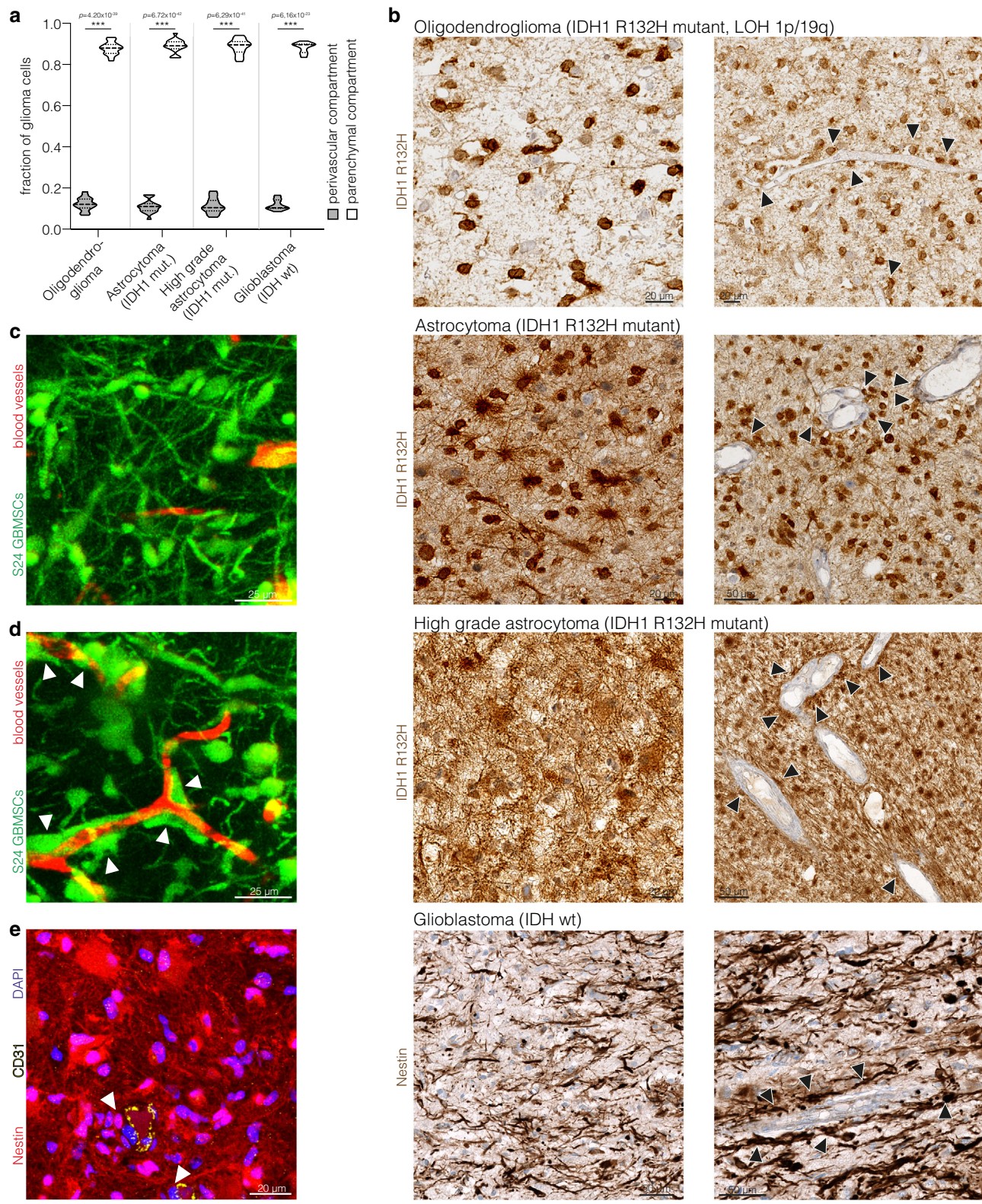

IDH1 mutant high grade astrocytoma ($p = 0.0005$), and IDH wild-type glioblastoma ($p = 0.0064$). In contrast, in oligodendroglioma, the distribution of proliferating cells matched the general distribution of tumor cells in the different compartments ($p = 0.2339$, two-tailed $t$-tests), arguing against a specific quiescence of cells in the PVN in this very glioma entity. In summary, perivascular cells constitute a major fraction of long-term resident astrocytoma and glioblastoma cells, both at the invasive front and in the established tumor core, and are characterized by a quiescent state.

**PVN location, tumor cell networks, and glioma resilience.** In neurogenic niches and brain tumors alike, the PVN provides a specific microenvironment for stem-like cells, which has been

**Fig. 1 Perivascular and network niches in human and experimental gliomas. a** Distribution of IDH1 R132H positive glioma cells and nestin positive glioblastoma cells in the perivascular and parenchymal compartment in molecularly defined patient glioma specimen (oligodendroglioma: $n = 18$ patients; astrocytoma: $n = 19$ patients; high grade astrocytoma: $n = 20$ patients; glioblastoma: $n = 10$ patients; one-way ANOVA, Tukey's post hoc test). **b** Exemplary immunohistochemical stainings of IDH1 R132H in molecularly defined human glioma specimen and of nestin in human glioblastoma specimen demonstrate the dense interconnection of glioma cells in astrocytoma and glioblastoma specimen (left column) as well as the close association of tumor cells with blood vessels (arrowheads) (right column). **c** Exemplary two photon microscopy (2-PM) image of highly interconnected S24 glioblastoma stem-like cells (GBMSCs) growing in the mouse brain. **d** Intravital 2-PM image of perivascular S24 GBMSCs (arrowheads) in a mouse xenograft. **e** Exemplary immunofluorescence image of a human glioblastoma specimen (glioblastoma, IDH wild-type, ATRX retained, MGMT promoter unmethylated) demonstrates the dense interconnection of nestin-positive glioma cells and their relation to the perivascular niche (CD31-positive endothelium). Arrowheads exemplarily mark perivascular tumor cells. Data **a** are represented as violin plot with median and quartiles. (mut.: mutant; LOH: loss of heterozygosity; wt: wild-type). ***$p < 0.001$. Source data are provided as a Source Data file.

---

linked to cellular resilience[6,7,12,24]. Furthermore, our group recently characterized the integration of glioma cells into TM-connected multicellular networks as a mechanism of resistance to cytotoxic therapies and prerequisite for tumor self-repair[15,18]. This leads to the question of the interrelation and relative relevance of both key factors of glioma progression and tumor cell persistence. More cells in the parenchymal compartment extended two TMs (Fig. 4a), a phenotype associated with fast invasion[17]. Moreover, a significant fraction of cells without any TMs was only found in the perivascular compartment (Fig. 4a). The degree of integration into multicellular networks did not appear to be relevantly biologically different between parenchymal and perivascular tumor cells though (Fig. 4b).

After application of radiotherapy, long-term in vivo microscopy revealed that PVN cells are highly radioresistant, whereas the number of parenchymal cells was significantly reduced by therapeutic irradiation as early as 7 days later (Fig. 4c, d). To rule out that the higher resistance of PVN cells was due to their sole integration into the TM-connected tumor network, a subgroup analysis was performed (Fig. 4e). Interestingly, even non-connected cells in the PVN were largely radioresistant, in contrast to the non-connected cells in the parenchyma which significantly regressed after radiotherapy. As expected[15], TM-connected cells resisted the cytotoxic effects of radiotherapy, irrespective of their compartment (Fig. 4e).

Likewise, after temozolomide (TMZ) chemotherapy, a higher resistance of PVN glioblastoma cells was confirmed in the S24 GBMSC line, and again this resistance could not be explained by differential tumor cell connectivity, at least not solely (Fig. 4f). In T269 gliomas derived from another MGMT promoter hypermethylated GBMSC line which is more sensitive to chemotherapy than S24 and also more quickly proliferating in vivo[18], perivascular glioblastoma cells also resisted TMZ better than parenchymal ones (Fig. 4g).

After surgery and also after targeted ablation of one single tumor cell within the tumor cell network, a repair response with directed extension of newly formed TMs towards the lesion, replacement of network-integrated glioblastoma cells, and repopulation of the damaged tumor region has been described[18,25]. We hence investigated whether perivascular and parenchymal tumor cells differed in their reactivity to a direct traumatic event. After inducing a microtrauma by ablation of one single glioblastoma cell with a high-power laser beam, an early reaction of a perivascular cell in this tumor microregion very frequently occurred within two hours, while parenchymal cells showed less reactivity (Fig. 5a). Of note, this early reaction was independent of the TM network connectivity of the ablated cell ($p = 0.119$, Fisher's exact test), thereby suggesting a paracrine rather than a network-mediated signal. Furthermore, after milder laser damage to a larger tumor area (Fig. 5b) as well as after a surgical lesion (Fig. 5c), a marked increase in tumor cell count in the damaged area was reproducibly inducible. This "malignant healing response" was strikingly accentuated in the PVN

(Fig. 5b–d) and preceded the dense repopulation of the lesioned brain area (arrowheads in Fig. 5c–e).

In summary, the perivascular localization of glioblastoma cells was strongly associated with their therapy resistance and resilience, additive to TM connectivity. Despite their static and often dormant behavior under resting conditions, perivascular cells can exhibit a strong reactivity to damage and initiate a dense repopulation of lesioned brain tumor regions.

**PVN colonization and TM networks are inversely regulated by NOTCH1.** Next we asked the question how both niches of resistance are molecularly interrelated: can glioblastoma cells dynamically change their position between the PVN and TM network niche, cross-compensating a molecular deficiency that deprives them of one or the other? For this we first inhibited network function by knockdown of the gap junction protein Cx43, which is a known connector of TM networks[15]. Here, morphological tumor cell networks greatly decreased, and tumor cells were driven into a perivascular position with high numbers of perivascular tumor cell clusters (Fig. 6a).

This led to the question whether a defect in PVN colonization in turn drives TM network integration of glioblastoma cells. Several publications suggested that activation of the NOTCH1 pathway is associated with glioma cell properties in the PVN, including cellular stemness[7,26,27]. Interestingly, the NOTCH1 pathway also plays an important role in neurite outgrowth[28], which shares many features with TM formation[15–17,29]. Hence, we sought to elucidate its relevance for both the PVN and TM network niche. Immunohistological analyses demonstrated a NOTCH1 pathway activation in perivascular tumor cells (Fig. 6b), confirming previous results[7].

We next used shRNA mediated knockdown of NOTCH1 to investigate the effects on brain and niche colonization as well as tumor cell morphology in vivo. Downregulation of NOTCH1 expression led to decreased vascular co-option (Fig. 6c) and a reduction of the perivascular tumor cell population (Fig. 6d). On the other hand, TM formation was markedly induced, with longer TMs (Fig. 6e), an early shift to highly interconnected cellular subpopulation with more than 4 TMs (Fig. 6f), and the formation of dense multicellular networks (Fig. 6g, h).

Brain tumor cells integrated into multicellular networks and those remaining non-connected in vivo can be separated based on their differential uptake of the fluorescent dye SR101[15,21]. We separated both cell populations by FACS and performed RNA sequencing of bulk populations. Analyses of the differential expression in two primary glioblastoma cell lines (S24 and T269) confirmed downregulation of the NOTCH1 receptor and of several genes associated with NOTCH1 pathway activation in the connected tumor cell population, and also upregulation of genes associated with NOTCH1 pathway inhibition (Table 1).

In contrast to glioblastomas, 1p/19q-codeleted oligodendrogliomas are characterized by TM network deficiency[15] (Fig. 7a).

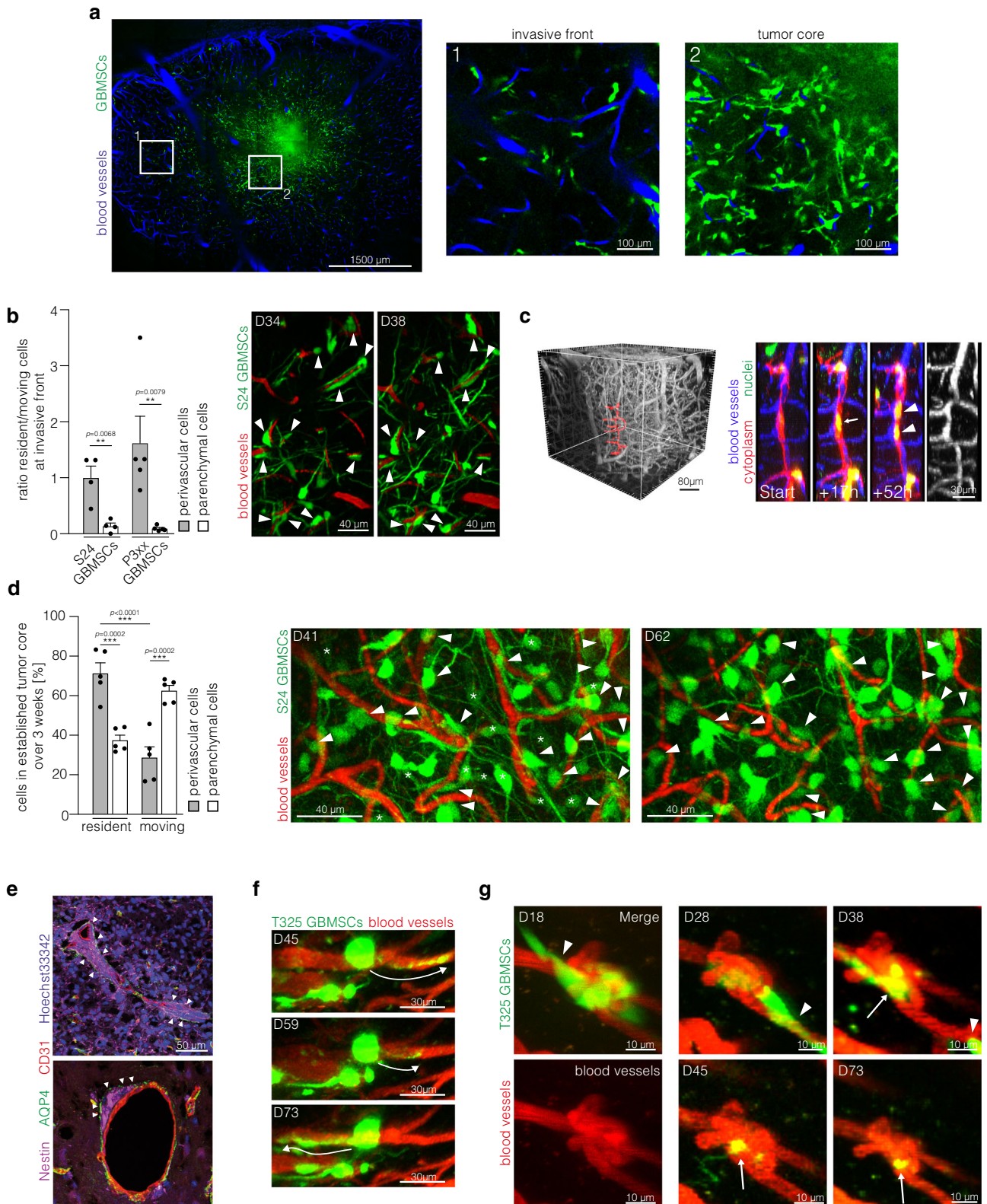

Compared to the TM- and network proficient S24 glioblastoma cells, BT088 oligodendroglioma cells show a higher *NOTCH1* expression (Fig. 7a), in line with the suggested role of *NOTCH1* for TM network biology. Likewise, by analyzing TCGA gene expression data of molecular glioblastomas vs. oligodendrogliomas[15], we found a relative downregulation of the *NOTCH1* gene and several downstream targets of the *NOTCH1* pathway in glioblastoma, as well as an upregulation of genes associated with *NOTCH1* pathway inhibition (Table 2). When compared to astrocytoma and glioblastoma cells, oligodendroglioma cells showed a comparable perivascular niche colonization in the xenograft model (Fig. 7b) and in human oligodendroglioma specimen (Fig. 1a). Finally, after shRNA-mediated knockdown of *NOTCH1*, BT088 oligodendroglioma cells were not able to give rise to tumors in immunodeficient

**Fig. 2 Long-term glioma cell behavior in the perivascular niche. a** Overview intravital 2-PM image of the tumor bearing hemisphere (S24 glioblastoma stem-like cells (GBMSCs), D23). Right: Magnifications show the invasive front (1) and the tumor core (2). **b** Left: Ratio of resident and moving S24 and P3xx GBMSCs at the invasive front over 4 (S24) and 5 (P3xx) days (*n* = 4 regions (S24; 41–86 cells per region)/5 regions (P3xx; 23–57 cells per region) in 3 mice per cell line, two-tailed *t*-test (S24) and two-tailed Mann Whitney test (P3xx)). Right: Exemplary intravital 2-PM images of the same region at the invasive front imaged over 4 days demonstrating several resident perivascular cells (arrowheads) (S24 GBMSCs, D34-38). **c** Left: 3D in vivo 2-PM image of the vasculature. The vessels shown on the right side are marked in red. Right: Timeseries of resident and dividing S24 GBMSC (arrow) in the perivascular niche over 52 h. Daughter nuclei are indicated with arrowheads. The angiogram is shown on the right. Blood vessels (blue), nuclei (green), cytoplasm (red). **d** Left: Analysis of residency and migration in the established tumor core over 21 days (D41–62) (S24 GBMSCs, in vivo 2-PM, *n* = 5 regions in three mice, 46–81 cells per region, one-way ANOVA ($p = 5.36 \times 10^{-6}$), Tukey's post hoc test for multiple comparisons). Right: Exemplary intravital 2-PM images of the same region at day 41 and 62. Resident perivascular S24 GBMSCs are marked with arrowheads, parenchymal cells moving away are marked by asterisks. **e** Immunofluorescence staining of nestin (purple), Aquaporin 4 (AQP4) (green), CD31 (red), and Hoechst33342 (blue) demonstrates the localization of perivascular cell clusters (arrowheads) below the astrocyte endfeet (S24 GBMSCs). **f** Exemplary in vivo 2-PM images of T325 GBMSCs over 28 days showing a resident tumor cell extending and retracting TMs (arrows). **g** Repetitive 2-PM imaging of T325 GBMSCs (green) demonstrating a resident cell (arrow) associated with a remodeled blood vessel (red). Some cells from the initial cell cluster migrated away during the observation period (arrowheads). Data **b**, **d** are represented as mean + SEM. **\*\***\*p* < 0.01, **\*\*\****p* < 0.001. Source data are provided as a Source Data file.

mice as cells slowly regressed over time (Fig. 7c). All in all, this data further support the importance of *NOTCH1* as a regulator of TM network connectivity, but also suggests *NOTCH1* as a principal regulator of oligodendroglioma growth, without specifically affecting PVN biology in this glioma subgroup.

**_NOTCH1_-dependent niche positions regulate growth and resistance in gliomas**. To understand how these differential changes of tumor cell niche integrations after *NOTCH1* down-regulation influence the overall sensitivity of 1p/19q intact gliomas to cytotoxic therapy, we analyzed tumor cell responses to radio-therapy, and found it decreased for the total shNOTCH1 glio-blastoma cell population (Fig. 8a). Deeper analysis revealed that this can exclusively be attributed to the increased radioresistance of the more abundantly TM-interconnected shNOTCH1 cells, whereas the fewer non-TM connected cells were even more vul-nerable to radiotherapy in shNOTCH1 tumors (Fig. 8b). Further subgroup analysis revealed that this effect was due to an increased therapy response of the remaining non-connected perivascular cells (Fig. 8c), suggesting that *NOTCH1* proficiency is relevant for the resistance of perivascular cells to radiotherapy. In vitro, *NOTCH1* downregulation inhibited proliferation of glioblastoma cells (Fig. 8d); in vivo, tumor bearing mice showed an improved survival when *NOTCH1* was downregulated (Fig. 8e). High-field MRI confirmed the brain tumor growth attenuation after *NOTCH1* knockdown (Fig. 8f, g). In contrast, after irradiation, the small and growth-deficient brain tumors were highly therapy resistant, with no regression being detectable compared to sham treatment (Fig. 8f, g). In summary, *NOTCH1* proficiency plays an important role for the colonization of the PVN and tumor growth, whereas its deficiency results in impaired growth but also induces TM- and network formation that makes the tumor more radioresistant (Fig. 8h).

## Discussion

The dynamics, fate, and relevance of individual glioma cells in the perivascular niche was revealed in this study over weeks and months. In line with previous work investigating tumor cell dynamics in different tumor areas[30], tumor cell invasion was not restricted to the invasive front but also prevalent in the dense tumor core. Here, we show that these dynamics are characteristic for parenchymal cells, whereas a majority of perivascular cells shows residency, even at the invasive front. The go-or-grow hypothesis suggests that glioma cells either migrate or pro-liferate[31]. Our data suggest that the invading parenchymal cells constitute the proliferating subpopulation, which is in line with recent single cell analyses[32], arguing for a go-and-grow (and not

go-or-grow) mechanism that should be further investigated in future studies. A minority of the resident perivascular cells was closely associated with remodeled blood vessels forming capillary loops or glomeroloid bodies[33], further suggesting reciprocity in the perivascular niche[8,10]. Previous studies demonstrated that glomeroloid bodies are populated by stem-like tumor cells in human tumor specimen[34] and this microvascular pattern is associated with a worse outcome in patients[35,36].

During neurogenesis, the PVN provides a specific micro-environment for neural stem cells[37]. A similar PVN for cancer stem-like cells has been proposed for primary brain tumors[6,7,10,12] and these stem-like cells have been associated with therapy resistance and tumor recurrence[2,6,7,38]. Recently, a metabolic zonation and consequent cellular heterogeneity dependent on the distance from blood vessels have been demonstrated in gliomas and perivascular cells exhibited a more aggressive phenotype and pronounced therapy resistance[39]. Our data confirm that tumor cells in the perivascular region are highly resistant against cyto-toxic therapies, largely independent of the integration into mul-ticellular networks. If this resistance is due to the slower proliferation rate, metabolic differences and/or specific micro-environmental cues, such as *NOTCH1* signaling discussed below, remains to be elucidated. Here, despite its residency, perivascular glioma cells demonstrated reactiveness to nearby damage includ-ing surgical lesions by inducing an early gliotic reaction and a subsequent (re-)population of the lesioned brain area, thereby sharing traits with neural stem cells[40] and reactive astrocytes, which also possess stem cell features[41].

Endothelial cells are known to activate *NOTCH1* signaling in adjacent glioma cells, including glioma stem-like cells[4,7,8,26]. *NOTCH1* inhibitors have been tried in various preclinical studies, partially demonstrating effects on stem-like cells, although con-flicting data exists and evidence for efficacy in clinical trials is still lacking[42,43]. Making use of the shRNA mediated knockdown of *NOTCH1* we provide one possible explanation for those con-flicting results: Although downregulation of *NOTCH1* led to a depletion of the perivascular cell population, reduced tumor growth and reduced therapy resistance of cells in the PVN in line with previous reports[7,26,44,45], it also induced TM- and network formation, thereby rendering the growth-deficient tumors vir-tually unsusceptible for the cytotoxic effects of radiotherapy. *NOTCH1* proficiency is hence a key mechanism for the resilience of cells in the PVN, firstly by allowing population of that niche and secondly by being prerequisite for the protective properties of this specific microenvironment. In line with our results outlined here, many publications demonstrate a growth-inhibitory effect of *NOTCH1* downregulation[42,46–50]. The differences regarding response to cytotoxic therapies[46–50] might be explained by the

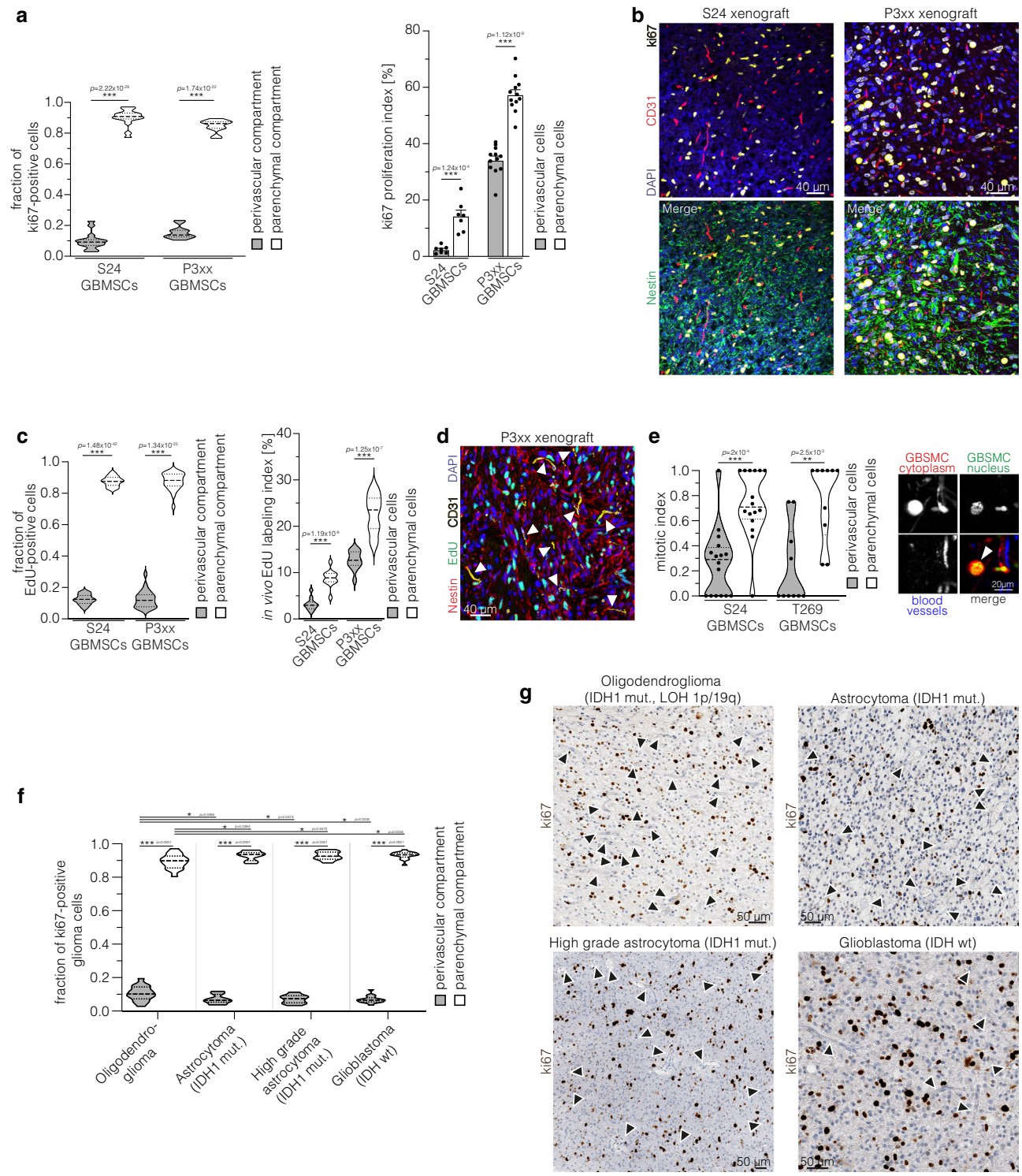

different models used and their TM-proficient phenotype. Our data suggest that *NOTCH1* is an important mediator of the resistance of perivascular tumor cells and the resistance promoting effects of *NOTCH1* downregulation can be attributed to the induction of TMs. The latter might not be reflected in all models: especially cultivation under serum-containing conditions, which were used for many studies on *NOTCH1* in glioblastoma[49], leads to TM-deficient tumors[17].

The formation of TMs shares many features with axon and neurite outgrowth, which is underlined by the phenotype observed after *NOTCH1* knockdown. During neurodevelopment, *NOTCH1*

activation inhibits neurite sprouting[28,51], whereas *NOTCH1* inhibition has been shown to promote neurite extension[52]. *NOTCH1* deficiency is only the third known molecular driver of TMs. Despite the caveats in targeting this pathway, this can provide insights into the molecular biology of multicellular networks and how to target them[16,53,54].

Our data suggests that *NOTCH1* inhibition might be used to slow down tumor growth but should be avoided in combination with cytotoxic therapies due to the induction of tumor cell networks. In TM-deficient oligodendrogliomas, we found an upregulation of the *NOTCH1* pathway, both in the BT088 cell line and

**Fig. 3 Perivascular cells are slow-cycling. a** Left: Distribution of ki67-positive cells in the perivascular and parenchymal compartment in orthotopic S24 ($n = 16$ regions from three mice, two-tailed $t$-test) and P3xx ($n = 12$ regions from four mice, two-tailed $t$-test) xenograft tumors. Right: ki67 proliferation index (ki67 positive cells/all cells in the respective compartment) of S24 ($n = 3$ mice, two-tailed $t$-test) and P3xx tumor cells ($n = 4$ mice, two-tailed $t$-test) in the perivascular and the parenchymal compartment in xenograft tumors. **b** Exemplary immunofluorescence stainings of ki67 (yellow, proliferating cells), nestin (green, tumor cells), DAPI (blue, nuclei), and CD31 (red, endothelium) in S24 and P3xx xenografts. **c** Left: Quantification of the distribution of EdU-positive tumor cells (indicating cells in S-phase) after in vivo incorporation of EdU (4 h) in the parenchymal and perivascular compartment (S24 xenografts, $n = 25$ regions from 5 mice, two-tailed $t$-test; P3xx xenografts, $n = 12$ regions from 4 mice, two-tailed $t$-test). Right: EdU labeling index (EdU positive cells/all cells in the respective compartment) in the perivascular and the parenchymal compartment (S24 xenografts, $n = 5$ mice, two-tailed $t$-test; P3xx xenografts, $n = 4$ mice, two-tailed $t$-test). **d** Exemplary immunofluorescence staining of EdU (green, cells in S-phase), nestin (red, tumor cells), DAPI (blue, nuclei) and CD31 (yellow, endothelium) demonstrates that most proliferating cells are located in the parenchymal compartment (P3xx xenograft). Blood vessels are marked with arrowheads. **e** Mitotic index (number of mitotic events normalized to the distribution of cells in the perivascular and parenchymal compartment) of S24 and T269 GBMSCs ($n = 16$ regions in eight mice (S24), $n = 10$ regions in 4 mice (T269), two-tailed Mann Whitney tests). Right: Exemplary in vivo 2-PM of mitosis of a parenchymal S24 GBMSC (arrowhead). Cytoplasm (red), nucleus (green), blood vessels (blue). **f** Distribution of ki67 positive cells in the parenchymal and perivascular compartment in molecularly defined human glioma specimen (oligodendroglioma: $n = 16$ patients; astrocytoma: $n = 19$ patients; high grade astrocytoma: $n = 15$ patients; glioblastoma: $n = 16$ patients; one-way ANOVA ($p = 1.36 \times 10^{-139}$), Tukey's post hoc test for multiple comparisons). **g** Exemplary immunohistochemical stainings of ki67 in molecularly defined human glioma specimen. Arrowheads exemplarily mark blood vessels. Data in violin plots **a**, **c**, **e**, **f** are represented as median and quartiles. Data in columns **a** are represented as mean + SEM. \*$p < 0.05$\*\*, $p < 0.01$, \*\*\*$p < 0.001$. (mut: mutant; LOH: loss of heterozygosity; wt: wild-type). Source data are provided as a Source Data file.

human samples, which is in line with the principal regulation of TM networks by *NOTCH1*. However, our analyzes also show fundamental biological differences between 1p/19q-codel oligodendrogliomas and 1p/19q-intact other glioma types: in contrast to astrocytomas and glioblastomas, the *NOTCH1* pathway activation in oligodendroglioma seems not to be strongly related to PVN position and cells in the PVN were not slow cycling.

Thus, our data imply that at least two, partially additive and complementary, niches of cellular resilience exist in glioblastoma: a PVN and a TM connectivity niche, that inversely depend on *NOTCH1* expression. Phenotypic adaptation[55] or switches between both niches must be considered for targeted therapies.

In conclusion, the perivascular compartment accommodates the majority of long-term resident glioma cells and is associated with resistance against cytotoxic therapies. Downregulation of the *NOTCH1* pathway depletes this cell pool and diminishes the protective properties of the perivascular niche, but concomitantly induces TM- and network formation with subsequent therapy resistance. All in all, this describes two niches of progression and resistance in gliomas—the TM connectivity niche and the PVN—and speaks for a complex and partially complementary role of these two. To effectively treat incurable astrocytomas and glioblastomas, novel concepts to target both niches seem of high importance, ideally avoiding cross-compensatory tumor cell reactions.

## Methods

**Cell lines.** The patient-derived glioblastoma cell lines S24, T269, T325 and P3xx and BT088 oligodendroglioma cells were kept under stem-like neurosphere conditions (37 °C, 5.0% $CO_2$): DMEM F-12 medium (31330-038, Invitrogen), B27 supplement (12587-010, Invitrogen), 5 µg ml⁻¹ insulin (I9278, Sigma-Aldrich), 5 µg ml⁻¹ heparin (H4784, Sigma-Aldrich), 20 ng ml⁻¹ epidermal growth factor (rhEGF; 236-EG, R&D Systems), and 20 ng ml⁻¹ basic fibroblast growth factor (bFGF; PHG0021, Thermo Fisher Scientific). S24 (human, female), T269 (human, male), T325 (human, male), and P3xx (human) were authentificated (Multiplexion GmbH, Germany). S24, T269, and T325 were further authenticated as glioblastoma by Illumina 850k methylation array[56]. BT088 cells were obtained from ATCC (ATCC CRL-3417, RRID:CVCL_N708) (human, male). The molecular characterization of all used cancer cell lines can be found in Supplementary Tables S1 and S2.

**Lentiviral transductions.** For transduction, cells were incubated with lentiviral particles and 10 µg ml⁻¹ polybrene (TR-1003-G, Merck Millipore) for 24 h. Cell lines were stably transduced with the LeGO-T2 vector (cytoplasmatic tdTomato, Addgene 27342, RRID:Addgene_27342), pLKO.1-LV-GFP (H2B-GFP, Addgene #25999, RRID: Addgene_25999), pLenti6.2 hygro/V5-Lifeact-YFP, plKO.1-puro-CMV-TurboGFP_shnon-target (cytoplasmatic GFP, SHC016, Sigma-Aldrich), plKO.1-puro-CMV-TurboGFP_shCX43 (*CX43* knockdown, sequence: GCCCAAACTGATGGTG TCAA, Sigma-Aldrich), and plKO.1-puro-CMV-TurboGFP_shNOTCH1 (*NOTCH1* knockdown, sequence: CCGGGACATCACGGATCATAT, Sigma-Aldrich).

*NOTCH1* and *CX43* knockdown were confirmed by Western blot analysis (45% (S24 GBMSCs); 92% (BT088) (*NOTCH1*) / 80% (S24 GBMSCs) (*CX43*) reduction in protein expression). Anti-Notch1 (1:1000; #4380, Cell Signaling Technology, RRID: AB_10691684) and anti-GAPDH antibody (1:5000; LAH1064, Linaris) were used for Western blot analyses.

**Animal procedures.** 8-to-10-week-old male NMRI nude mice (Charles River) were used for studying the intracranial tumor growth and dynamics. Mice were kept in individual housing after cranial window implantation (constant housing conditions: temperature $22 \pm 2$ °C, humidity $55 \pm 10\%$, 12 h light/dark cycles). All animal procedures were performed in accordance with the institutional laboratory animal research guidelines after approval of the responsible animal welfare officer (German Cancer Research Center, Heidelberg, Germany) and the regional council (Referat 35, Regierungspräsidium Karlsruhe, Germany). A chronic cranial window and a titan ring that allows pain-free fixation of the animals during two photon microscopy were implanted as described before[15,17]. Two weeks after window implantation 30,000 patient-derived and fluorescently labeled glioblastoma stem-like cells suspended in PBS were intracranially implanted in a depth of 500 µm. For survival and MRI studies, 50,000 cells (S24 shControl and shNotch1) were implanted without preparation of a chronic cranial window. T2-weighted rapid acquisition with refocused echoes sequence MRI images were acquired on a 9.4 T horizontal bore MR scanner (BioSpec 94/20 USR, Bruker BioSpin) with a four-channel phased-array surface coil (parameters: TE = 33 ms, TR = 2500 ms, flip angle = 90°, acquisition matrix: $200 \times 150$, number of averages = 2, slice thickness = 700 µm duration = 2 min 53 s). For radiotherapy experiments, whole brain irradiation with fractions of 7 Gy at a dose rate of 3 Gy min⁻¹ were performed on 3 consecutive days at D60 ($\pm 10$ days) using a 6 MV linear accelerator (Artiste, Siemens) with a 6 mm collimator adjusted to the window size. As control no radiation was applied (sham radiation). For chemotherapy experiments, mice were treated with 100 mg/kg body weight temozolomide for 3 consecutive days on D85 $\pm 3$ after tumor implantation by oral gavage. For the surgical lesion experiment, an established tumor region was chosen by two photon microscopy, the cranial window was removed and a 26-gauge Hamilton syringe was used to resect a cylindrical volume within the established tumor region as described before[18]. The perilesional tumor regions were followed repetitively by two photon microscopy afterwards. For two photon microscopy and MRI imaging mice were anesthetized with isoflurane. Mice were scored clinically and rapidly killed if they showed neurological symptoms or a weight loss of >20%.

**In vivo multiphoton laser scanning microscopy.** Intravital two photon microscopy (2-PM) was performed with a Zeiss 7MP microscope (Zeiss) equipped with a Coherent Chameleon UltraII laser (Coherent) on anesthetized mice (4% isoflurane for initiation, 0.5–2% for maintenance of anesthesia). A custom-made aperture allowed painless fixation of the head for imaging. Body temperature was permanently controlled by a rectal thermometer and was kept constant by a heating pad. Angiograms allowed identification of the same tumor regions over time. For angiograms, fluorescent dextranes (10 mg ml⁻¹, TRITC-dextrane (500 kDa; 52194, Sigma Aldrich) or FITC-dextrane (2000 kDa, FD2000S, Sigma Aldrich)) were applied by tail vein injection. The following wavelengths were used for excitation of different fluorophores: 750 nm (FITC-dextrane, tdTomato), 850 nm (GFP, TRITC-dextrane), and 950 nm (tdTomato, YFP). Appropriate filter sets (band pass 500–550 nm/band pass 575–610 nm) were used. For ablation of single glioma cells, the laser beam was focused on the GFP-labeled nucleus (H2B-GFP) and continuous scanning was performed until disintegration of the cell. For damage of a

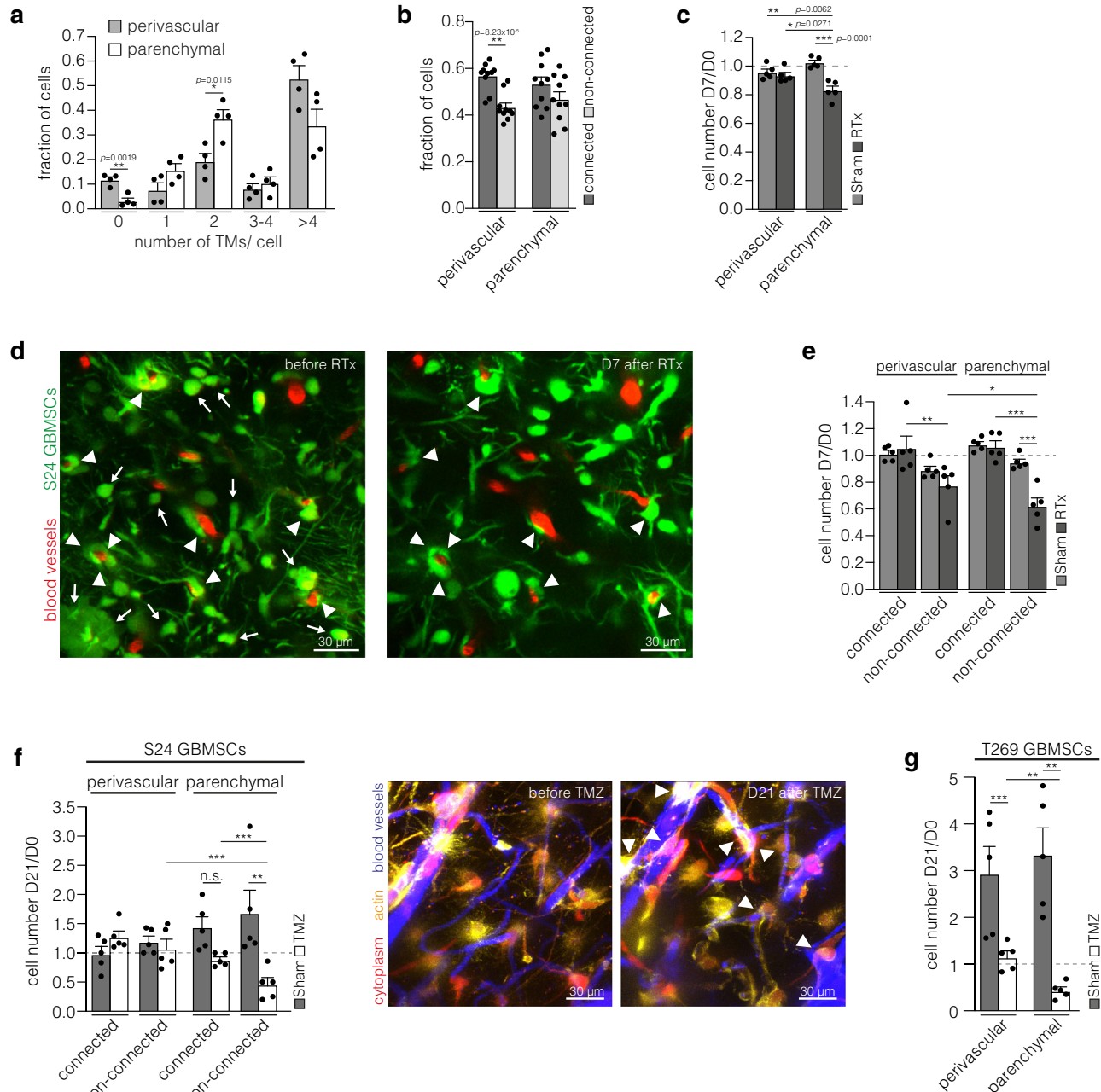

**Fig. 4 Both perivascular and network glioma cells are particularly resilient. a** Analysis of fractions of cells with different number of TMs (S24 glioblastoma stem-like cells (GBMSCs), D41, $n = 4$ regions in 3 mice, 52–104 cells, two-tailed $t$-tests). **b** Analysis of the connectivity of cells in the perivascular and parenchymal compartment (S24 GBMSCs, D65 ± 4, $n = 10$ regions in 6 mice, 148–259 cells, two-tailed $t$-tests). **c** Ratio of tumor cell counts in the same regions 7 days after and before sham treatment or irradiation (RTx) (S24 GBMSCs, $n = 5$ regions in 3 mice per group, one-way ANOVA ($p = 0.0002$), Tukey's post hoc test). **d** Representative 2-PM images of the same tumor region before and 7 days after radiotherapy demonstrate that perivascular glioma cells are largely radioresistant (S24 GBMSCs). Arrows: disappearing parenchymal tumor cells. Arrowheads: surviving perivascular tumor cells. **e** Ratio of tumor cell counts in the same regions 7 days after and before sham treatment or irradiation (RTx) categorized by connectivity and compartment (S24 GBMSCs, $n = 5$ regions in 3 mice per group, one-way ANOVA ($p = 1.55; \times 10^{-6}$), Student–Newman–Keuls post hoc test). The analysis revealed that non-connected tumor cells in the parenchyma are most susceptible for the cytotoxic effects of radiotherapy. In contrast, non-connected cells in the PVN are more radioresistant, thereby demonstrating the protective environment promoted by the PVN. In the PVN, interconnection has an additive effect on radioresistance. **f** Left: Ratio of cell counts of S24 GBMSCs in the same regions 21 days after and before temozolomide or sham treatment ($n = 5$ regions in 3 mice per group, one-way ANOVA on ranks ($p = 0.0018$), Student–Newman–Keuls post hoc test). Right: Exemplary 2-PM images before and 21 days after temozolomide treatment in S24 GBMSCs. Arrowheads: surviving perivascular GBMSCs. **g** Ratio of cell counts of T269 GBMSCs in the same regions 21 days after and before temozolomide or sham treatment ($n = 5$ regions in 3 mice per group, one-way ANOVA on ranks ($p = 0.0002$), Student–Newman–Keuls post hoc test). Data **a–c**, **e–g** are represented as mean + SEM. *$p < 0.05$, **$p < 0.01$, ***$p < 0.001$, n.s. = not significant. Source data are provided as a Source Data file.

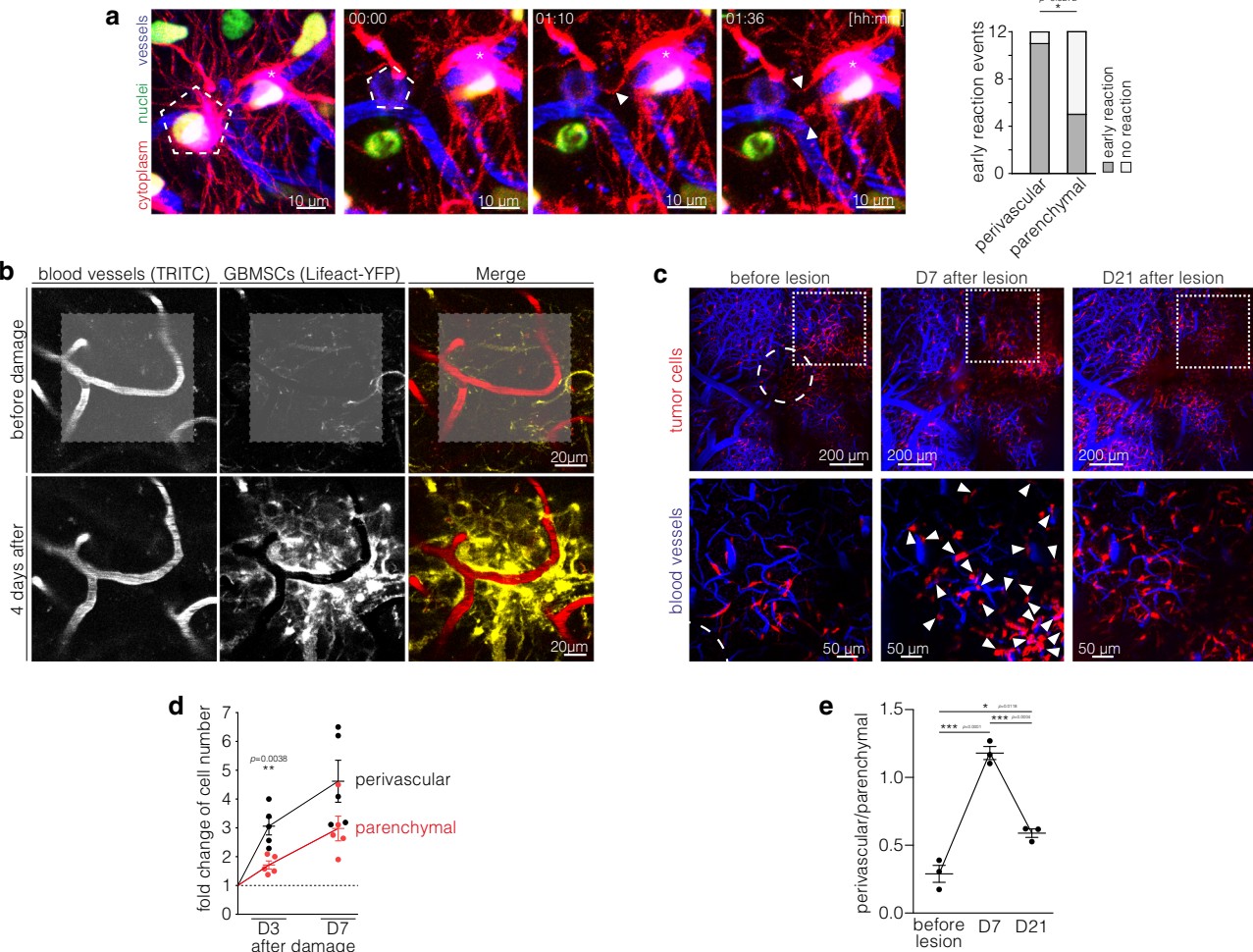

**Fig. 5 Perivascular cells govern glioma's damage response. a** Left: Early reaction (white arrowheads) of a perivascular cell (asterisk) after laser ablation of a nearby S24 GBMSC (pentagon). Cytoplasm (red), blood vessels (blue), nuclei (green). Right: Analysis of the early reaction to laser ablation of nearby cells (S24 glioblastoma stem-like cells (GBMSCs), $n = 12$ events, two-sided Fisher exact test). **b** After laser-induced stress (marked with gray area in the upper row) a strong perivascular reaction can be observed (S24 GBMSCs). **c** Representative in vivo 2-PM images of the same region before, 7 and 14 days after a surgical lesion (circle) in a S24 xenograft tumor. 7 days after lesioning there is a strong increase of the perivascular cell population (arrowheads). Upper row: overview; lower row: subset. Subset regions are marked by dotted squares. **d** Fold changes of perivascular (black) and parenchymal (red) cell counts 3 and 7 days after laser damage compared to before laser damage (S24 GBMSCs, $n = 5$ regions in 4 mice, two-tailed $t$-tests) indicate that the damage response is predominated by the PVN. **e** Quantification of the ratio of perivascular and parenchymal cells before, 7 and 14 days after a surgical lesion (S24 GBMSCs, $n = 3$ animals, one-way ANOVA ($p = 3.92 \times 10^{-5}$), Tukey's post hoc test). This analysis reveals a shift towards the perivascular compartment that is most pronounced in the initial phase and precedes the repopulation of the lesioned brain area. Data **d**, **e** are represented as mean ± SEM. *$p < 0.05$, **$p < 0.01$, ***$p < 0.001$. Source data are provided as a Source Data file.

larger tumor area, repetitive scanning (950 nm wavelength) for ~8 min with high laser power was performed. For the evaluation of therapeutic responses, the same tumor regions were imaged repetitively over 1–3 weeks. For invasion speed measurements and identification of invading tumor cells, single glioma cells were followed with repetitive imaging over 24 h. Short-term residency of individual glioma cells was determined by repetitive imaging over 4 (S24) and 5 days (P3xx), long-term residency by repetitive imaging between D41 and D62.

**SR101 staining and separation of connected and non-connected tumor cells**. For separation of connected and non-connected tumor cells, mice bearing S24 and T269 GFP tumors were injected with sterilized saline solution dissolving SR101 (S359, Invitrogen) i.p. at a dose of 0.12 mg per gram b[57]ody weight, which is taken up by tumor cells within 5–8 h. Mice were then perfused with PBS and the brains were harvested. Whole brain single cell suspensions were prepared with brain tumor dissociation kit (130-095-942, Miltenyi Biotec) and gentleMACSTM Dissociator (Miltenyi Biotec). After dissociation the suspension was resuspended in FACS buffer (PBS + 1% FCS, 10500064, ThermoFisher). FACS was performed on a FACSAria cell sorter (BD Biosystems). Cells were stained with Calcein Violet 450 AM (65-0854-39, Invitrogen) and TO-PRO®-3 Iodide (T3605, Invitrogen) for 10 min on ice before sorting for the viable cell population (Calcein Violet 450high and TO-PRO-3neg). Within the viable cell population SR101^high, GFP^high (connected

tumor cells), and SR101^low, GFP^low (non-connected tumor cells) were separated. The YG586/15 channel was used to visualize SR101 signal. FACS was performed as part of another study[57]. The exemplary visualization of the gating strategy provided in Supplementary Fig. S1b was adopted from this study[57].

**Bulk RNA sequencing of connected and non-connected tumor cells**. After cell sorting, cell populations were lysed (lysis buffer, RNeasy Micro Kit, 74004, Qiagen). mRNA was isolated and purified in accordance with the manufacturer's instruction. RNA was pooled from 3 mice for each sample. The conversion of RNA to dsDNA was done with the SMARTer Ultra Low Input RNA for Illumina Sequencing (Clontech), the libraries were then prepared using NEBNext® ChIP-Seq Library Prep Master Mix Set for Illumina (E6240, New England Biolabs) and sequenced on HiSeq2000 v4 (Illumina) in 50 bp single-end mode by our core facility. The quality of bases was evaluated using the FASTX Toolkit. Homertools 4.7 were applied for PolyA-tail trimming[58]; reads with a length of <17 were removed. The filtered reads were mapped with STAR 2.3[59] against the human reference genome (GRCh38) and PicardTools 1.78 with CollectRNASeqMetrics were used for quality checking. Count data were generated by htseq-count using the gencode.v26.annotation.gtf file for annotation[60]. DESeq2 1.4.1 was run with default parameters for the group-wise comparison[61]. The expression levels were transformed to logarithmic space using log2.

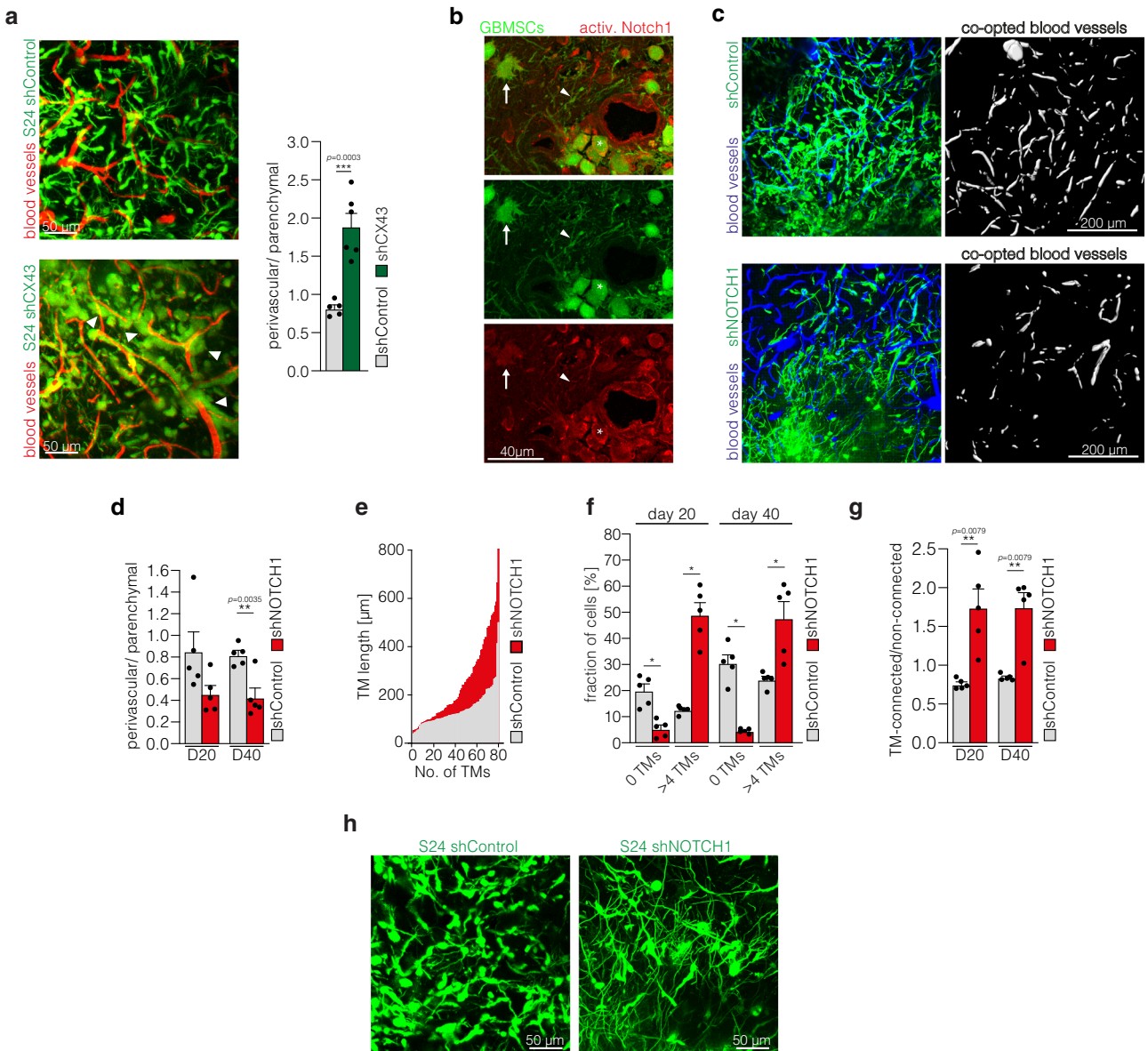

**Fig. 6 NOTCH1 inhibition leads to depletion of the perivascular niche but promotes TM-network formation. a** Left: Exemplary in vivo 2-PM images of S24 shControl (top, D40) and shCX43 tumors (bottom, D43). Right: Corresponding ratio of perivascular and parenchymal cells in S24 shControl and shCX43 knockdown tumors (D40± 4, $n = 5$ regions (S24 shControl)/6 regions (S24 shCX43) in 3 mice per group, two-tailed $t$-test). **b** Immunofluorescence staining of activated Notch1 in a S24 tumor section. Arrow: multi-TM parenchymal cell (low Notch1 activation); arrowhead: Notch1-low TM; asterisk: perivascular cell (high Notch1 activation). **c** Exemplary 2-PM images of S24 shControl (top) and shNOTCH1 (bottom) glioblastoma stem-like cells (GBMSCs) (green) with segmentation of co-opted blood vessels (white) demonstrating a reduced population of the perivascular compartment after *NOTCH1* downregulation (D24 ± 1). Blood vessels (blue). **d** Ratio of the perivascular and parenchymal cell count 20 and 40 days after S24 shControl and shNOTCH1 tumor cell implantation ($n = 5$ regions in 3 mice (S24 shControl)/4 mice (S24 shNOTCH1), 38–445 cells, two-tailed $t$-tests). **e** Histogram and quantification of TM length in S24 shControl and shNOTCH1 GBSMSCs (D20, $n = 3$ mice (S24 shControl), $n = 4$ mice (S24 shNOTCH1). **f** Quantification of TM-devoid (0 TMs) vs. TM-rich (>4 TMs) S24 shControl and shNOTCH1 GBMSCs ($n = 5$ regions in 3 mice (S24 shControl)/4 mice (S24 shNOTCH1), 18–286 cells, one-way ANOVA on the ranks ($p = 2.90 \times 10^{-12}$), Student–Newman–Keuls post hoc test). **g** Ratio of connected and non-connected S24 shControl and shNOTCH1 GBMSCs 20 and 40 days after tumor cell implantation ($n = 5$ regions in 3 mice (S24 shControl)/4 mice (S24 shNOTCH1), 11–245 cells, two-tailed Mann–Whitney tests). **h** Corresponding 2-PM images demonstrating the TM richness of S24 shNOTCH1 tumors (D40). Data **a, d, f, g** are represented as mean + SEM. *$p < 0.05$, **$p < 0.01$, ***$p < 0.001$. Source data are provided as a Source Data file.

**Analysis of differential mRNA expression of human gliomas**. RNA sequencing data of molecular (1p/19q-codeleted, IDH mutant) oligodendroglioma ($n = 56$) and molecular (1p/19q non-codeleted, IDH wild-type) glioblastoma ($n = 70$) was downloaded from TCGA and analyzed as described previously[15]. The full list of differentially expressed genes can be found in Osswald et al. (2015)[15].

**In vivo EdU incorporation**. Animals were treated with 50 mg/kg EdU i.p. (BCK488-IV-IM-S, baseclick GmbH) 4 h prior to euthanasia. Animals were

cardially perfused with 4.5% phosphate-buffered formaldehyde solution (Roti-Histofix 4.5%, 2213, Carl Roth). Tissue was frozen after cryoprotection with 30% sucrose solution and 10 µm cryosections were cut. The sections were permeabilized with 0.5% Triton X-100 in PBS for 20 min. The EdU Click Kit (BCK488-IV-IM-S, baseclick GmbH) was used for EdU detection. After washing, sections were incubated with the reaction cocktail for 30 min as per the manufacturer's instructions. Afterwards sections were incubated with anti-nestin (1:400, ab6320, Abcam, RRID:AB_308832) and anti-CD31 (1:50; PA5-16301, ThermoFisher Scientific, RRID:AB_10981955) antibodies. Goat anti-rabbit Alexa Fluor 594 (1:400;

**Table 1 Interconnected tumor cells exhibit *NOTCH1* pathway downregulation.**

Bulk RNA seq. of connected vs. non-connected tumor cells in vivo

*NOTCH1* pathway activation

| Gene | log2 fold change (S24 GBMSCs) | Adjusted *p* value (S24 GBMSCs) | log2 fold change (T269 GBMSCs) | Adjusted *p* value (T269 GBMSCs) |
|---|---|---|---|---|
| NOTCH1 | −0.8411 | 0.0097 | −0.6468 | 0.0122 |
| HES6 | −1.4296 | 6.2899e−6 | −1.4490 | 5.4451e−8 |
| DLL1 | −2.3525 | 5.8021e−14 | −2.6558 | 4.9849e−28 |
| MFNG | −1.8398 | 1.018e−5 | −2.8051 | 1.3372e−12 |
| SATB1 | −1.8029 | 1.8930e−6 | −1.3847 | 1.9251e−8 |
| MYCL | −1.2588 | 0.0016 | −0.8474 | 0.0178 |
| MYCN | −1.1416 | 0.0378 | −1.0212 | 3.1898e−8 |
| CDKN1C | −1.2697 | 0.0010 | −2.3721 | 9.7218e−11 |
| SOX8 | −1.0591 | 1.3633e−4 | −1.5615 | 4.3253e−27 |
| ASCL1 | −0.9464 | 0.0067 | −1.1870 | 0.0081 |
| PTPRJ | −0.8798 | 0.0350 | −1.5500 | 0.0021 |

*NOTCH1* pathway inhibition

| Gene | log2 fold change (S24 GBMSCs) | Adjusted *p* value (S24 GBMSCs) | log2 fold change (T269 GBMSCs) | Adjusted *p* value (T269 GBMSCs) |
|---|---|---|---|---|
| CDK1 | 0.8195 | 0.0076 | 1.2082 | 7.9934e−5 |
| CHI3L1 | 1.1131 | 3.7427e−4 | 1.5617 | 2.5644e−6 |

Comparison of differential expression of genes associated with *NOTCH1* pathway activation and inhibition respectively between connected and non-connected S24 and T269 orthotopically xenografted glioblastoma cells. Connected and nonconnected tumor cells were separated by FACS based on their uptake of SR101 in vivo before bulk RNA seq. was performed. Only genes with significant differential expression as determined by an adjusted *p* value < 0.05 are shown.

A-11037, ThermoFisher Scientific, RRID:AB_2534095) and goat anti-mouse Alexa Fluor 633 (1:400; A-21052, ThermoFisher Scientific, RRID:AB_2535719) secondary antibodies were used. Images were acquired using a Leica TCS SP5 confocal microscope (LAS software version 2.7.3.9723).

**Immunofluorescence and confocal microscopy.** For immunofluorescence of mouse tissue, tumor bearing mice were cardially perfused with PBS, followed by 4.5% phosphate-buffered formaldehyde solution (Roti-Histofix 4.5%, 2213, Carl Roth). Brain tissue was incubated in 30% sucrose solution overnight for cryoprotection and snap frozen afterwards. Heat-induced epitope retrieval with 0.01 M citrate buffer, pH 6.0, was performed. The following primary antibodies were used: anti-nestin (1:400, ab6320, Abcam, RRID:AB_308832), anti-CD31 (1:100, AF3628, R&D Systems, RRID:AB_2161028), anti-aquaporin 4 (1:200, ab9512, Abcam, RRID:AB_307299), anti-activated Notch1 (1:50, ab8925, Abcam, RRID: AB_306863) and anti-ki67 (1:500, ab15580, Abcam, RRID:AB_443209). Donkey anti-mouse IgG Alexa Fluor 488 (1:400; A-21202, Thermo Fisher Scientific, RRID: AB_141607), goat anti-mouse IgG Alexa Fluor 488 conjugate (1:400; A-11029, Thermo Fisher Scientific, RRID:AB_138404), donkey anti-goat IgG Alexa Fluor 633 (1:400; A-21082, Thermo Fisher Scientific, RRID:AB_141493) and donkey anti-rabbit IgG Alexa Fluor 546 (1:400; A-10040, Thermo Fisher Scientific, RRID: AB_2534016) secondary antibodies were used. Images were acquired using a Leica TCS SP5 confocal microscope.

**Immunohistochemical and immunofluorescence staining of human tumor specimen.** Tissue sections of resected primary gliomas were obtained from the Department of Neuropathology in Heidelberg in accordance with local ethical approval (Ethikkommission der Universität Heidelberg, Heidelberg University, Heidelberg, Germany). Samples were anonymized prior to analyzes and all patients gave their informed consent for the scientific use of resected tissue samples. For immunohistochemical nestin staining samples from ten patients from the Heidelberg Neuro-Oncology center diagnosed with glioblastoma, IDH wild-type based on histopathological and molecular characteristics were used. Inclusion of patients into this analysis is covered by a local Heidelberg ethics vote (No. S307/2019). 3 µm sections were incubated with the anti-nestin antibody, clone 10C2 (1:200; MAB5326, Merck Millipore, RRID:AB_11211837). Nestin expression was detected using the ultraView DAB protocol on the automated VENTANA BenchMark ULTRA platform (Roche, Switzerland). Ventana standard signal amplification and ultra-wash was followed by counterstaining with Hematoxylin II (790-2208, Roche, Switzerland) and blueing reagent (760-2037, Roche, Switzerland).

Immunohistochemistry of IDH1 R132H and ki67 was performed on 0.5-μm-thick formalin-fixed, paraffin-embedded (FFPE) tissue sections. For IDH1 R132H immunohistochemistry, sections were stained on a BenchMark XT immunostainer, for ki67, staining was performed on a BenchMark Ultra immunostainer (Ventana Medical Systems, Tucson, AZ, USA). Anti-IDH1 R132H (1:20; DIA-H09, Dianova, RRID:AB_2335716) and anti-Ki67 (1:100; M7240, Dako, RRID:AB_2142367)

antibodies were used respectively. Slides were scanned on an Aperio AT2 Slide Scanner (Biosystems Switzerland AG, Switzerland).

For immunofluorescence staining of nestin and CD31 in human glioblastoma specimen, 100 µm thick, PFA-fixed sections were used. Heat-induced antigen retrieval was performed with Tris-EDTA buffer. Sections were incubated with anti-nestin (1:400; ab6320, Abcam, RRID:AB_308832) and anti-CD31 (1:50; PA5-16301, ThermoFisher Scientific, RRID:AB_10981955) antibodies for 24 h. Goat anti-rabbit Alexa Fluor 594 (1:400; A-11037, ThermoFisher Scientific, RRID: AB_2534095) and goat anti-mouse Alexa Fluor 633 (1:400; A-21052, ThermoFisher Scientific, RRID:AB_2535719) secondary antibodies were used. Auto-fluorescence was quenched with Sudan Black.

**Tissue selection and molecular characterization of human tumor specimen.** Tumor tissues were selected based on their methylation data from the database of the Department of Neuropathology Heidelberg[62]. All methylation data were generated using the Illumina MethylationEPIC (850k) array platform according to the manufacturer's instructions (Illumina, San Diego, USA). Sample preparation was performed as previously described[63]. DNA methylation status of 10.000 CpG sites was analyzed on the current version v12b4 of the Classifier (https://www.molecularneuropathology.org/mnp). Of note, all cases presented a calibrated Classifier Score > 0.9.

The probability of MGMT promoter methylation from 850k array data was estimated as previously described[64]. For cases with an MGMT promoter methylation status not determinable by 850k methylation analysis, additional pyrosequencing was performed using the therascreen® MGMT Pyro® kit (QIAGEN®) and the PyroMark® Q24 system (QIAGEN®) according to the manufacturer's protocol. Bisulfite conversion was done with the EpiTect fast DNA bisulfite kit (QIAGEN®). In accordance with published studies[65–67], a mean MGMT promoter methylation percentage < 8% across the investigated CpG sites was considered as non-methylated and a value ≥8% was considered as methylated.

**Patient characteristics.** The series used for immunohistochemical analyses consisted of tissues with the following molecular characteristics:

1. Oligodendroglioma: *n* = 18 (for IDH1 R132H staining), *n* = 16 (for ki67 staining). Methylation class IDH glioma, subclass 1p/19q codeleted oligodendroglioma, all ATRX retained.

2. Astrocytoma (IDH1 R132H mutant): *n* = 19 (for IDH1 R132H staining), *n* = 19 (for ki67 staining). Methylation class IDH glioma, subclass astrocytoma, all ATRX loss, all 1p/19q non-codel.

3. High grade astrocytoma (IDH1 R132H mutant): *n* = 20 (for IDH1 R132H staining), *n* = 15 (for ki67 staining). Methylation class IDH glioma, subclass high grade astrocytoma, all 1p/19q non-codel. a) IDH1 R132H staining: 16 ATRX loss, 3 ATRX retained, 1 undetermined. b) ki67 staining: 11 ATRX loss, 3 ATRX retained, 1 n/a.

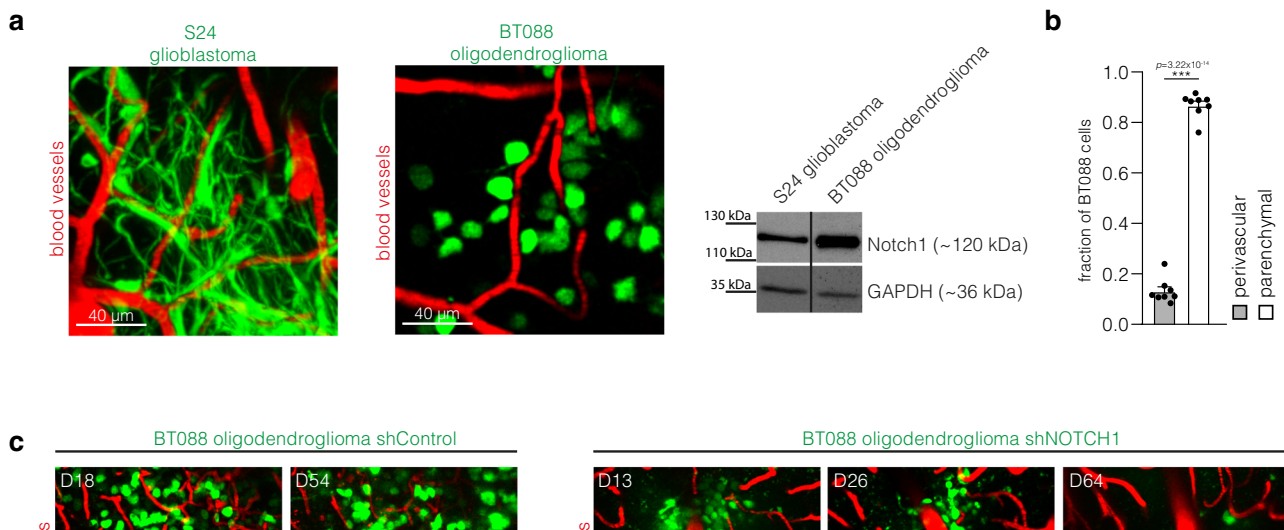

**Fig. 7 NOTCH1 pathway activation in TM-deficient oligodendrogliomas. a** Left: Exemplary intravital 2-PM images of S24 glioblastoma and BT088 oligodendroglioma cells (D21 after tumor implantation). Right: Western blot analysis of the Notch1 expression in S24 glioblastoma stem-like cells (GBMSCs) and BT088 oligodendroglioma cells. Loading control: GAPDH. **b** Quantification of the distribution of BT088 oligodendroglioma cells in the perivascular and parenchymal compartment ($n = 8$ regions in 5 mice, D21 ± 3, two-tailed $t$-test). **c** Exemplary images of BT088 shControl oligodendroglioma cells and BT088 shNOTCH1 cells demonstrate tumor regression after NOTCH1 knockdown. Data **b** are represented as mean + SEM. ***$p < 0.001$.

---

**Table 2 The NOTCH1 pathway is activated in human oligodendrogliomas compared to glioblastomas.**

**Molecular human glioblastoma vs. molecular human oligodendroglioma**

*NOTCH1* pathway activation

| Gene | Entrez ID | log2 fold change | *p* value | FDR |
|---|---|---|---|---|
| SOX8 | 30812 | −2.5180613 | 5.39E−43 | 5.67E−41 |
| DLL3 | 10683 | −2.4262425 | 1.26E−17 | 1.33E−16 |
| HEY1 | 23462 | −1.2942635 | 9.18E−23 | 1.53E−21 |
| MYC | 4609 | −1.2213157 | 3.76E−15 | 3.05E−14 |
| HES6 | 55502 | −1.1642784 | 1.29E−08 | 5.07E−08 |
| DLL1 | 28514 | −1.0985485 | 9.11E−08 | 3.25E−07 |
| SATB1 | 6304 | −0.8875692 | 5.68E−11 | 2.92E−10 |
| ZMIZ1 | 57178 | −0.8720752 | 7.95E−17 | 7.77E−16 |
| NOTCH1 | 4851 | −0.5352691 | 5.91E−05 | 0.00014523 |
| CDKN1B | 1027 | −0.4286673 | 2.34E−09 | 1.00E−08 |
| CCND1 | 595 | −0.3288129 | 0.12248947 | 0.16628677 |
| ASCL1 | 429 | −0.2774871 | 0.1584482 | 0.20949394 |
| CDKN1A | 1026 | −0.1732208 | 0.40000144 | 0.46701398 |
| PTPRJ | 5795 | −0.1423327 | 0.20323507 | 0.26025615 |
| JAG1 | 182 | 1.45417948 | 3.46E−21 | 4.94E−20 |

*NOTCH1* pathway inhibition

| Gene | Entrez ID | log2 fold change | *p* value | FDR |
|---|---|---|---|---|
| CHI3L1 | 1116 | 5.41391741 | 2.64E−35 | 1.46E−33 |
| NUMB | 8650 | 0.6442456 | 1.12E−19 | 1,46E−18 |

Comparison of RNA expression data (TCGA) of molecular glioblastoma ($n = 56$) and molecular 1p/19q-codeleted oligodendroglioma ($n = 70$) reveals downregulation of genes associated with *NOTCH1* pathway activation in glioblastoma and upregulation of genes associated with *NOTCH1* pathway inhibition.
*FDR* False discovery rate.

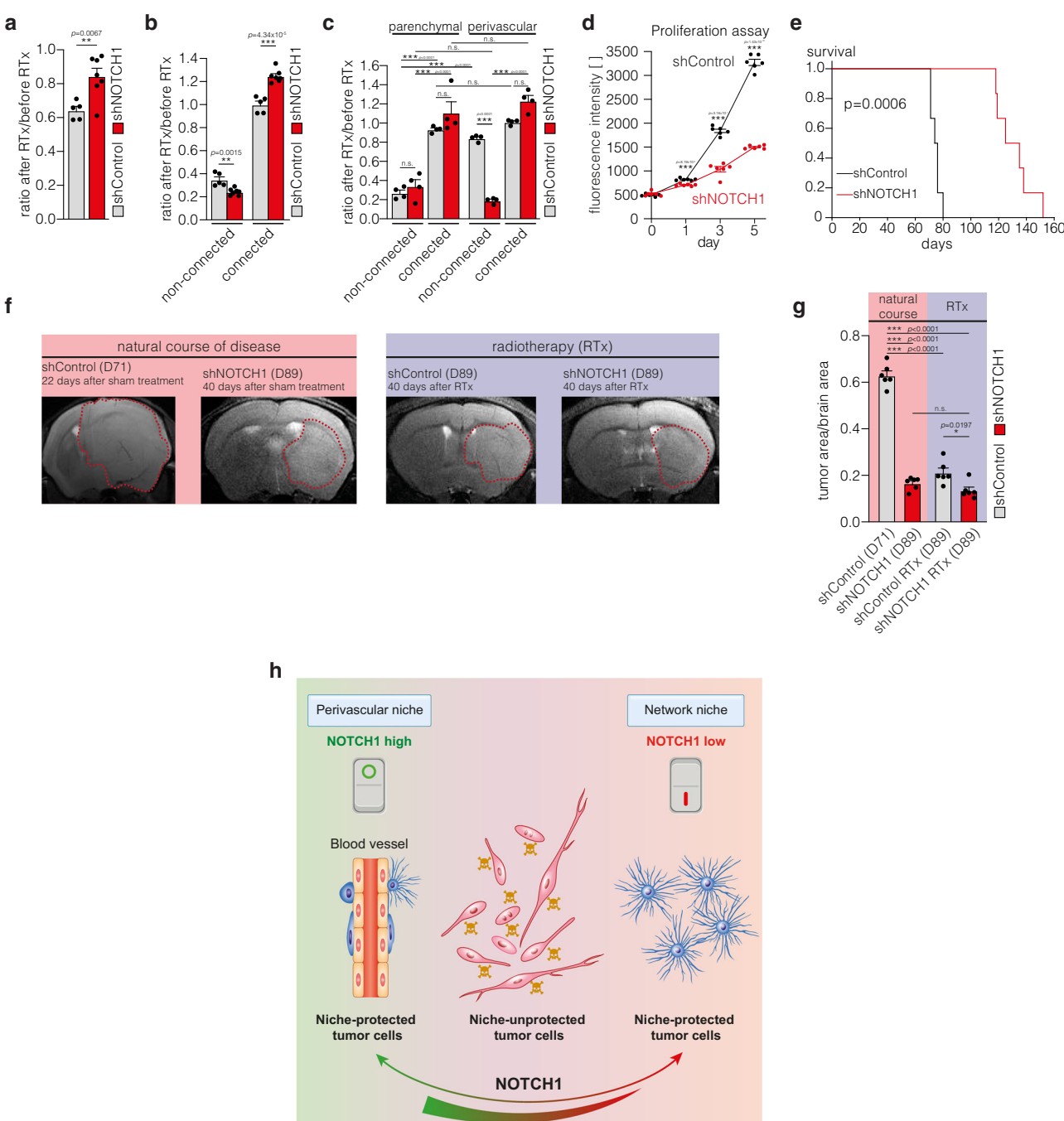

**Fig. 8 NOTCH1 downregulation sensitizes perivascular glioblastoma cells to radiotherapy, but induces network resistance. a, b** Ratio of cell counts 7 days after and before irradiation of S24 shControl and shNOTCH1 tumors ($n = 5$ regions (S24 shControl)/7 regions (S24 shNOTCH1) in 3 mice per group, two-tailed $t$-tests). **c** Subgroup analysis of the ratio of cell counts of S24 shControl and shNOTCH1 cells 7 days after and before irradiation categorized regarding connectivity and compartment ($n = 4$ regions in 3 mice per group, one-way ANOVA ($p = 9.09 \times 10^{-14}$), Tukey's post hoc test). **d** AlamarBlue proliferation assay of S24 shControl and shNOTCH1 cells ($n = 6$ biological replicates, two-tailed $t$-tests). **e** Survival analysis of S24 shControl and shNOTCH1 tumor bearing mice ($n = 6$ mice per group, two-sided log rank test). **f** Exemplary 9.4 T MRI images of S24 shControl and shNOTCH1 tumors. Left: natural course, right: 40 days after irradiation (RTx). **g** Quantification of tumor burden (tumor area/brain area on 9.4 T MRI) in untreated and irradiated mice ($n = 6$ mice per group, one-way ANOVA ($p = 6.18 \times 10^{-15}$), Tukey's post hoc test). **h** Graphical abstract of the two niches of resistance in glioblastoma. The perivascular niche (PVN) (left) is sustained by the NOTCH1 pathway, whereas low NOTCH1 expression leads to depletion of the perivascular niche and the induction of resistant multicellular networks (right). Non-connected, parenchymal cells (middle) in contrast are sensitive to cytotoxic therapies. Data **a–d**, **g** are represented as mean + SEM. *$p < 0.05$, **$p < 0.01$, ***$p < 0.001$, n.s. = not significant. Source data are provided as a Source Data file.

4. Glioblastoma (IDH1 wild-type): $n = 10$ (for nestin staining), $n = 16$ (for ki67 staining). Methylation class glioblastoma, IDH wild-type. a) Nestin staining: subclass: RTK I = 1, RTK II = 7, mesenchymal=1, MYCN = 1. All 1p/19q non-codel, all MGMT promoter unmethylated, all TP53 wild-type, 9 ATRX retained, 1 n/a. b) ki67 staining: subclass: RTK I = 4, RTK II = 9, mesenchymal=3. All 1p/19q non-codel, 6 MGMT promoter methylated, 10 MGMT promoter unmethylated.

**Alamar blue proliferation assay**. 1000 GBMSCs were seeded in 96-well plates and AlamarBlue Reagent (DAL1025, Thermo Fisher Scientific) was added on days 0, 1, 3, and 5. Fluorescence intensity (excitation: 560 nm; emission: 590 nm) was measured using a SpectraMax M5e microplate reader (Molecular Devices) after 3 h.

**Image processing**. For image analysis and figures, images were processed using ZEN software (Zeiss, RRID:SCR_018163) and Imaris (Bitlane, RRID:SCR_007370). Image calculation and filtering was performed to reduce background noise. For figures, 3D stacks are transformed into orthogonal 2D maximum intensity projections (MIP).

**Quantifications**. All quantifications were performed manually in ImageJ (version 1.53, NIH, RRID:SCR_003070) and Imaris (Bitlane, RRID:SCR_007370) in high resolution (1024 * 1024 pixels) 3D z-stacks. Immunohistochemical and immuno-fluorescence images were analyzed in Qupath[68], Aperio ImageScope software (v11.0.2.725, Aperio Technologies, USA) and ImageJ (version 1.53, NIH, RRID: SCR_003070). Cells were regarded perivascular if the whole cell body was directly associated with a blood vessel. For quantification of ki67 positive cells in human tumor specimen, cells were regarded perivascular if the cell was located within 10 μm from a blood vessel and was not separated by another cell body. TM length of individual cells was measured by tracking TMs in z-stacks (3D images). For the mitotic index, mitotic events were analyzed in H2B-GFP expressing cells. The mitotic events were normalized to the distribution of cells in the parenchymal and perivascular compartment in the respective region, thereby assuming equal fractions in both compartments. The early reaction after single ablation was defined as the directed extension of TMs towards the damaged area within 100 min. For the quantification of the tumor/brain area ratio on MRI images, the tumor and the whole-brain area were measured manually in the image with the largest tumor expansion.

**Statistics and reproducibility**. Statistical analyses were performed using SigmaPlot (version 14.0, Systat Software, RRID:SCR_003210) and Prism 8.4.1 (GraphPad, RRID: SCR_002798). Normal distribution of datasets was assessed with a Shapiro–Wilk test. Statistical significance of normally distributed data was assessed by a two-tailed Student's t test. Non-normally distributed data were assessed with a two-tailed Mann–Whitney rank-sum test. For datasets with >2 groups ANOVA or ANOVA on ranks with the appropriate post hoc tests (Tukey's or Student–Newman–Keuls tests) for multiple comparisons were performed. The $p < 0.05$ was considered statistically significant.

For ANOVA tests, $p$ values are reported in the figure legend, multiplicity adjusted $p$ values are noted in the figures and multiple test tables including confidence intervals can be found in the Source data file. No 95% confidence intervals for each difference or multiplicity adjusted exact $p$ values can be reported in case the Newman–Keuls post hoc test was used for multiple comparisons, as this test works in sequential fashion. Statistical details including group sizes can be found in the respective figure legends. All experiments were repeated at least three times unless otherwise noted.

**Reporting summary**. Further information on research design is available in the Nature Research Reporting Summary linked to this article.

## Data availability
The RNA seq. data of connected and non-connected glioblastoma cells have been deposited in the Sequence Read Archive (SRA) database under the accession number PRJNA554870. Uncropped Western blot data are provided in Source data and Supplementary Fig. S1a. Source data are provided with this paper.

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

## Acknowledgements
We thank Hrvoje Miletic for providing the P3 cell line. We thank Miriam Gömmel, Cathrin Löb, Chanté Mayer and Susann Wendler for technical assistance. We thank Matthia Karreman for critical reading of the manuscript. This study was funded by the Deutsche Forschungsgemeinschaft (DFG, German Research Foundation) – Project-ID 404521405, SFB 1389 – UNITE Glioblastoma, Work Packages A01 (E.J., F.W.), A03 (W.W., T.K., D.C.H.), A06 (F.S.), and D01 (M.R., A.v.D.), as well as DFG, KU 3555/1-1 (F.K.). F.S. is fellow of the Else Kröner Excellence Program of the Else Kröner-Fresenius Stiftung (EKFS).

## Author contributions
E.J., M.O., W.W. and F.W. designed the study, interpreted the data and wrote the manuscript. E.J. and M.O. analyzed data. E.J., M.O., M.R. and S.W. performed two photon microscopy imaging. R.X. performed FACS and RNA-seq. analyses. E.J. performed immunofluorescence stainings. F.K. and S.H. performed MRI imaging and interpreted the data. H.D., D.C.H., T.K., F.S. and A.v.D. performed and interpreted immunohistochemical stainings and molecular tissue analyses.

## Funding

## Competing interests
E.J., M.O., W.W. and F.W. report the patent (WO2017020982A1) "Agents for use in the treatment of glioma". F.W. is co-founder of DC Europa Ltd (a company trading under the name Divide & Conquer) that is developing new medicines for the treatment of glioma. Divide & Conquer also provides research funding to F.W.'s lab under a research collaboration agreement. All other authors declare no competing interests.
