## [Peer Review File · Nature Communications]

Reviewers' Comments:

Reviewer #1:

Remarks to the Author:

Review of Jung et al.

Using IDH1 mutant specific staining, GBM cells were seen near and far from vessels, forming a network connected by tumor microtubes (TMs). Using intravital imaging, perivascular niche (PVN) cells were much less motile. Proliferation of PVN glioma was significantly lower. Cells in the parenchyma not connected to other cells via TMs were more sensitive to radiation or temozolomide. Blood vessels were highly sensitive to local tumor cell ablation, and tumor regrowth was potentially increased around vessels. Knockdown of connexin43 reduced TM network formation and increased perivascular tumor cells. Knockdown of Notch1 reduced the PVN cells, and increased TM networks. Gene expression analysis indicated increased Notch signaling in connected cells. Compared to gliomas, oligodendroglioma cells show increased notch and notch related signaling. Downregulation of notch increased radioresistance of connected cells while non connected PVN cells were sensitive.

In summary this paper provides a possible explanation for the limited response of some GBMs to radio/chemotherapy, due to conversions between PVN and parenchymal connected cell populations which have different sensitivities. The combination of intravital imaging with chemotherapy and molecular manipulation is admirable. The presentation is a bit difficult to understand, perhaps because it follows the path of original experimentation rather than organization for clarity.

One question which I had regarding the conclusions was - why are oligos more sensitive to cytotoxic therapies if they have increased notch, just like PVN?

Other minor points to address

In figure 2G it seems that the cells associated with the vessel either moved away or died.

The interpretation of Figure 4D is complex – it appears that nonconnected cells in the parenchyma are more sensitive to radiation. It is unclear if any other comments conclusions can be drawn from Figure 4D – was ANOVA analysis performed?

Quantitation of Figure 4H would help support the role of the vasculature in tumor regrowth.

Jeff Segall

Reviewer #2:

Remarks to the Author:

This manuscript by Jung et al., relates to tumor cell plasticity, heterogeneity and therapeutic resistance within crucial microenvironmental niches in glioma. The work presented is topical and of interest to the brain tumor basic research and neuro-oncology communities. Overall the authors make broad claims that are not substantiated by the data provided. Please see detailed comments below.

1- The detailed characterization of the model systems used has not been provided, i.e., status of ATRX, TP53 and 1p19q codeletion together with mutant IDH1 status. These markers are currently used to accurately classify mutant IDH1 gliomas and this information is critical in order to distinguish if the phenomenon described is a general feature of mutant IDH1 gliomas or if it is specific to the different glioma subtypes which harbor mutant IDH1.

2- The association of glioma cells with the PVN has been described previously by several other groups, thus this is not considered novel.

3- The data shown in Figure 1, Panel D should also be shown at higher magnification to provide an indication of the PVN within the tumor mass. This is critical for the accurate interpretation of the data.

4- The experiments to evaluate motility/migration are very superficially presented and the quantification is not rigorous.

5- For the quantification shown in Figure 3, panels A and B, the authors need to show immunocytochemistry images at lower magnification.

6- The authors should perform in vivo cell proliferation to conclusively demonstrate the mitotic status of glioma cells in the different niches.

6- The experiments using Notch1 knock down are not considered novel.

Response to reviewer comments

REVIEWER COMMENTS

Reviewer #1 (Remarks to the Author):

Review of Jung et al.

Using IDH1 mutant specific staining, GBM cells were seen near and far from vessels, forming a network connected by tumor microtubes (TMs). Using intravital imaging, perivascular niche (PVN) cells were much less motile. Proliferation of PVN glioma was significantly lower. Cells in the parenchyma not connected to other cells via TMs were more sensitive to radiation or temozolomide. Blood vessels were highly sensitive to local tumor cell ablation, and tumor regrowth was potentially increased around vessels. Knockdown of connexin43 reduced TM network formation and increased perivascular tumor cells. Knockdown of Notch1 reduced the PVN cells, and increased TM networks. Gene expression analysis indicated increased Notch signaling in connected cells. Compared to gliomas, oligodendroglioma cells show increased notch and notch related signaling. Downregulation of notch increased radioresistance of connected cells while non connected PVN cells were sensitive.

In summary this paper provides a possible explanation for the limited response of some GBMs to radio/chemotherapy, due to conversions between PVN and parenchymal connected cell populations which have different sensitivities. The combination of intravital imaging with chemotherapy and molecular manipulation is admirable. The presentation is a bit difficult to understand, perhaps because it follows the path of original experimentation rather than organization for clarity.

- We thank the reviewer for the positive and constructive feedback. We edited the manuscript to clarify (see page 5, line 77ff.) that we first characterized general features of glioma cells in the different compartments before then investigating how they respond to therapies and ultimately studying the effects of NOTCH1 inhibition on the aforementioned aspects (e.g. niche population, TM extension, therapy resistance, tumor growth). We have also tried fundamentally other presentation orders but feel that the one of the revised manuscript should still be most comprehensible and understandable. By providing much more patient data (new Fig. 1a,b,d, new Fig. 3f,g), the manuscript starts now with more proof of the fundamental fact of the existence of this niche in the human disease, which might also make things clearer.

One question which I had regarding the conclusions was - why are oligos more sensitive to cytotoxic therapies if they have increased notch, just like PVN?

- We appreciate the question of the reviewer. To investigate if the Notch1 pathway activation found in oligodendroglioma is driven by increased population of the PVN, we now looked into the distribution of oligodendroglioma cells in our BT088 oligodendroglioma mouse model (new Fig. 7b) as well as in human tumor specimen (new Fig. 1a). Here we found a comparable population of the PVN like in glioblastoma. Hence Notch1 pathway activation in oligodendroglioma seems not related to PVN position (as seen in astrocytoma and glioblastoma). The therapeutic sensitivity of oligodendroglioma can thus, among other factors, be explained by TM network deficiency as well the lack of a compensatory population of the PVN. Thus, NOTCH1 pathway activation alone seems not to be sufficient to render cells resistant, but our results suggest that it is prerequisite for population of the PVN, but only for astrocytomas and glioblastomas, which then promotes therapy resistance in these glioma entities. We have adopted the manuscript accordingly (page 10, line 212 ff.; page 14, line 313 ff.).

Other minor points to address

In figure 2G it seems that the cells associated with the vessel either moved away or died.

- It is true that some cells moved away from the remodeled blood vessel, while a single cell remained associated with it over weeks. We clarified this in the manuscript (page 6, line 99). In the figure, we marked cells that moved away as well as the resident cell.

The interpretation of Figure 4D is complex – it appears that nonconnected cells in the parenchyma are more sensitive to radiation. It is unclear if any other comments conclusions can be drawn from Figure 4D – was ANOVA analysis performed?

- We thank the reviewer for this remark. Indeed, an ANOVA analysis was performed. We have added this information to the figure legend, and edited the manuscript to make the important conclusions drawn from this quantification clearer (page 7, line 139 ff.). The two main conclusions are that non-connected tumor cells in the parenchyma are most susceptible for the cytotoxic effects of radiotherapy. In contrast, non-connected cells in the PVN are more radioresistant, hence demonstrating the protective environment promoted by the PVN. Furthermore, as connected cells in the PVN are more resistant than non-connected cells, interconnection in the PVN has an additive effect rendering cells fully resistant against radiotherapy. For better illustration, we have also added 2-photon microscopy images of the same tumor microregion before and after radiotherapy (new Fig. 4d) to illustrate the particular resistance of perivascular tumor cells.

Quantitation of Figure 4H would help support the role of the vasculature in tumor regrowth.

- We appreciate this interesting and clinically relevant suggestion. We performed two further experiments and corresponding quantitative analyses, which underlined the role of the PVN in the repair response and tumor regrowth after damage (new Fig. 5c-e). We not only analyzed the perivascular repair response after laser stress as suggested, but also looked into the response after a surgical lesion, which is the third standard of care treatment for glioblastoma (besides radio-chemotherapy). In both experiments, we found that a strong perivascular response precedes the repopulation of the resected tumor area, hence supporting the role of the vasculature for tumor relapse at the resection margin (see page 8, line 164 ff.).

Jeff Segall

Reviewer #2 (Remarks to the Author):

This manuscript by Jung et al., relates to tumor cell plasticity, heterogeneity and therapeutic resistance within crucial microenvironmental niches in glioma. The work presented is topical and of interest to the brain tumor basic research and neuro-oncology communities. Overall the authors make broad claims that are not substantiated by the data provided. Please see detailed comments below.

1- The detailed characterization of the model systems used has not been provided, i.e., status of ATRX, TP53 and 1p19q codeletion together with mutant IDH1 status. These markers are currently used to accurately classify mutant IDH1 gliomas and this information is critical in order to distinguish if the phenomenon described is a general feature of mutant IDH1 gliomas or if it is specific to the different glioma subtypes which harbor mutant IDH1.

- We appreciate and fully agree with the remark of the reviewer as it is indeed crucial to distinguish if the described features are specific for certain glioma entities. We now amended molecular details of all cell lines used (see Suppl. Table 1). In addition, we newly analyzed molecularly defined series of patients with different glioma subtypes (oligodendroglioma, IDH mutant astrocytoma, IDH mutant high-grade astrocytoma, and IDH wild type glioblastoma) for the distribution in the different niches as well as for ki67 analyses (new Fig. 1a,b and Fig. 3f,g). Here, we found that distribution in the parenchymal and perivascular compartment are similar in the different glioma subtypes. Moreover, the proliferation rate of perivascular cells is lower compared to the parenchymal cells in astrocytoma/glioblastoma. In contrast, in human oligodendroglioma, the ki67 positive fraction matched the fraction of cells in the PVN, hence arguing for similar proliferation rates in both compartments (see page 6, line 115 ff.). All in all, this data supports the notion that fundamental biological differences exist between 1p/19q codeleted vs intact gliomas, but similarities exist in the latter group of gliomas, thus independent of the IDH1 mutation (see page 14, line 313 ff.). Detailed patient characteristics can now also be found in the material and methods section (page 23, line 522 ff.).

2- The association of glioma cells with the PVN has been described previously by several other groups, thus this is not considered novel.

- We agree that the general concept of the PVN has been widely and thoroughly described, and studies investigating it in glioblastoma have been referenced in the introduction and discussion sections accordingly. In our study, we made use of longitudinal 2-photon microscopy to study single tumor cell dynamics and characteristics in the perivascular and the parenchymal compartment during natural course of disease (such as residency and mitotic activity) and in response to all three therapies currently considered standard of care (radio-chemotherapy and surgery). This is indeed novel, as is the investigation of its inter-relationship with the TM-connected tumor cell networks. To our knowledge similar intravital and longitudinal (over up to several weeks) studies on a single cell level have not been performed to date. We substantiate our findings by data from human tumor specimen (histological analyses in the new Fig. 1a,b, 3f,g as well as in Table 2). In addition, we used patient derived tumor cell lines which closely model the human disease, as demonstrated e.g. in the new Fig. 1b-d. We regard the heterogenous growth pattern prerequisite to understand cross-compensatory mechanisms. All in all, we here elucidate the interrelation and partial cross-compensation of the perivascular niche and the network niche for therapy resistance for the very first time. The network niche has only recently been described in glioblastoma and was not investigated in previous studies.

3- The data shown in Figure 1, Panel D should also be shown at higher magnification to provide an indication of the PVN within the tumor mass. This is critical for the accurate interpretation of the data.

- We thank the reviewer for this remark. The original image showed an IDH1 R132H staining, which did not allow to distinguish the PVN. We replaced it by a co-staining of nestin (tumor cells) and CD31 (endothelium) in a human glioblastoma specimen (new Fig. 1e) to demonstrate the interconnection of tumor cells as well as the population of the PVN. The image supplements the lower magnification IHC image shown in the new Fig. 1b.

4- *The experiments to evaluate motility/migration are very superficially presented and the quantification is not rigorous.*

- Motility/migration is an excellent and very interesting point indeed. We appreciate that the reviewer scrutinizes the data regarding residency and motility of perivascular and parenchymal tumor cells, and agree that it is only superficially presented here. In this current work we wanted to focus on long-term residency in the different compartment rather than on migratory properties of different cell populations. Repetitive 2-photon microscopy of the same tumor regions enables us to identify and track single cells over time, hence we regard the conclusions drawn valid.

[REDACTED]

5- *For the quantification shown in Figure 3, panels A and B, the authors need to show immunocytochemistry images at lower magnification.*

- We thank the reviewer for this remark. Based on the constructive criticism of the reviewer mentioned in point 5 and 1 we have now included a low magnification image of the immunofluorescence staining (new Fig. 3b) as well as IHC overview images from human tumor specimen (new Fig. 3g) with a corresponding quantification (new Fig. 3f).

6- *The authors should perform in vivo cell proliferation to conclusively demonstrate the mitotic status of glioma cells in the different niches.*

- We thank the reviewer for pointing out that the mitotic status of glioma cells in the different compartments needed further characterization. To determine the proliferating cell population (G1-, S- and G2-phase), we added additional ki67 stainings in P3xx xenografts (new Fig. 3a). As per the suggestion of the reviewer, we furthermore performed *in vivo* EdU incorporation in mice as a supplementary method indicating the cell population in S-phase (new Fig. 3c,d). These analyses confirmed that perivascular cells are slow cycling (see page 6, line 103 ff.). In addition, we now analyzed human tumor specimen to validate the results from our xenograft models (new Fig. 3f, g). Here we found that in astrocytoma and glioblastoma specimen, the majority of proliferating tumor cells is found in the parenchyma. When comparing the proliferating (ki67-positive) fraction and the distribution of cells in the different compartments by IDH1 R132H and nestin stainings in human tumor specimen, we found supporting data that perivascular tumor cells proliferate less (page 6, line 111 ff.).

6- *The experiments using Notch1 knock down are not considered novel.*

- We appreciate the concern of the reviewer. Notch1 knockdowns have indeed been studied in different models of glioma and Notch1 is known to be important for the perivascular niche. Here, we provide a novel explanation for the conflicting data on the effects of Notch1 down regulation on therapy resistance and limited clinical efficacy of therapies targeting the Notch1 pathway. In this study we identify the Notch1 pathway as a novel driver of TM outgrowth and network formation, which has not been reported before. Besides Gap43 and Ttyh1, the Notch1 pathway is only the third known driver of glioma cell interconnection and is therefore of interest for the development of TM-targeting strategies which is currently ongoing (Nature Biotechnology: <https://doi.org/10.1038/s41587-020-0411-9>). We show that Notch1 down-regulation can indeed have opposing and cross-compensatory effects on different niches of resistance (PVN and network niche), which complements what is known about the role of Notch1 on the PVN. All in all, we consider the discovery of the role of NOTCH1 as a “switch” between both prime niches of resistance in this disease a fundamentally novel and important finding that is first described in this manuscript. We have edited the manuscript to highlight how our results expand beyond what is already known (page 14, line 306 ff.).

[REDACTED]

Reviewers' Comments:

Reviewer #1:

Remarks to the Author:

The revisions in general are good. However, the additions on page 14 lines 306ff are confusing in that on lines 302 - 303 Notch1 is a positive regulator of TTYH1 which in turn is labeled as a driver of TMs. However, then line 306 describes NOTCH deficiency as a driver of TMs. Thus both notch increase and notch decrease are presented as driving TMs. I believe the authors are trying to discuss invasive, Notch induced TMs vs TMs that are resistant to Notch KD and perhaps are more stable cell to cell connections. However, this is very difficult to follow without reading other papers such as reference 17. I would recommend either removing all mention of TTYH1 because it is simply confusing or else a more extended discussion clarifying this apparent paradox (which seems to reflect the possible presence of two types of TMs which are not yet easily distinguishable at the molecular level).

Reviewer #2:

Remarks to the Author:

This manuscript by Jung et al, investigates the roles played by the tumor cell plasticity, heterogeneity and resistance in crucial microenvironmental niches in glioma. The authors have adequately addressed my previous comments. Their experimental designs are robust, the data is of high quality and the conclusions are in line with the data reported. Thus, this constitutes a novel and impactful manuscript.

Response to the reviewers' comments

Reviewer #1 (Remarks to the Author):

The revisions in general are good. However, the additions on page 14 lines 306ff are confusing in that on lines 302 - 303 Notch1 is a positive regulator of TTYH1 which in turn is labeled as a driver of TMs. However, then line 306 describes NOTCH deficiency as a driver of TMs. Thus both notch increase and notch decrease are presented as driving TMs. I believe the authors are trying to discuss invasive, Notch induced TMs vs TMs that are resistant to Notch KD and perhaps are more stable cell to cell connections. However, this is very difficult to follow without reading other papers such as reference 17. I would recommend either removing all mention of TTYH1 because it is simply confusing or else a more extended discussion clarifying this apparent paradox (which seems to reflect the possible presence of two types of TMs which are not yet easily distinguishable at the molecular level).

- We thank the reviewer for pointing this out. We have removed the respective paragraph for clarity.

Reviewer #2 (Remarks to the Author):

This manuscript by Jung et al, investigates the roles played by the tumor cell plasticity, heterogeneity and resistance in crucial microenvironmental niches in glioma. The authors have adequately addressed my previous comments. Their experimental designs are robust, the data is of high quality and the conclusions are in line with the data reported. Thus, this constitutes a novel and impactful manuscript.

- We thank both reviewers for their constructive feedback throughout the review process.